# The importance of spatial language for early numerical development in preschool: Going beyond verbal number skills

**Carrie Georges**[1]*, **Véronique Cornu**[2], **Christine Schiltz**[1]

**1** Department of Behavioural and Cognitive Sciences, Faculty of Humanities, Education and Social Sciences, University of Luxembourg, Esch-Belval, Luxembourg, **2** Centre pour le Développement des Apprentissages Grande-Duchesse Maria Teresa, Ministère de l'Éducation Nationale, de l'Enfance et de la Jeunesse, Strassen, Luxembourg

\* carrie.georges@uni.lu

**Data Availability Statement:** The data is available at the following link: https://osf.io/vg36r/ with the following identifier: DOI 10.17605/OSF.IO/VG36R.

**Funding:** This study is funded by the Luxembourg National Research Fund (FNR, https://www.fnr.lu/)

## Abstract

Recent evidence suggests that spatial language in preschool positively affects the development of verbal number skills, as indexed by aggregated performances on counting and number naming tasks. We firstly aimed to specify whether spatial language (the knowledge of locative prepositions) significantly relates to both of these measures. In addition, we assessed whether the predictive value of spatial language extends beyond verbal number skills to numerical subdomains without explicit verbal component, such as number writing, symbolic magnitude classifications, ordinal judgments and numerosity comparisons. To determine the unique contributions of spatial language to these numerical skills, we controlled in our regression analyses for intrinsic and extrinsic spatial abilities, phonological awareness as well as age, socioeconomic status and home language. With respect to verbal number skills, it appeared that spatial language uniquely predicted forward and backward counting but not number naming, which was significantly affected only by phonological awareness. Regarding numerical tasks that do not contain explicit verbal components, spatial language did not relate to number writing or numerosity comparisons. Conversely, it explained unique variance in symbolic magnitude classifications and was the only predictor of ordinal judgments. These findings thus highlight the importance of spatial language for early numerical development beyond verbal number skills and suggest that the knowledge of spatial terms is especially relevant for processing cardinal and ordinal relations between symbolic numbers. Promoting spatial language in preschool might thus be an interesting avenue for fostering the acquisition of these symbolic numerical skills prior to formal schooling.

## Introduction

Basic numerical skills are critically important for later mathematical development [1, 2] and have greater predictive value than reading or attention skills for later success in other academic

under the grant INTER/FWO/20/14716314 –
SpaNuMaDev. The funders had no role in study
design, data collection and analysis, decision to
publish, or preparation of the manuscript.

**Competing interests:** The authors have declared
that no competing interests exist.

domains [3]. Consequently, gaining better knowledge of the determinants of early numerical
development seems highly relevant both at the individual and societal level.

Early number knowledge is usually acquired prior to or outside of the school setting and
generally refers to the processing of nonsymbolic and symbolic (i.e., number words, Arabic
digits) number with respect to their cardinality or ordinality. According to the triple-code
model [4], this knowledge relies on three distinct numerical codes, serving different functions
and having distinct functional neurological bases, namely the analogical magnitude code, the
auditory verbal code, and the visual Arabic code. The analogical magnitude module represents
nonsymbolic quantities analogically over a spatially oriented mental number line, independent
of language. It underlies automatic access to approximate quantities in numerosity comparison
tasks. The verbal module represents numerical information lexically, phonologically and syn-
tactically. It is required amongst others for verbal counting, with the verbally rehearsed count-
list also supposedly underlying the early understanding of numerical order [5]. The visual Ara-
bic module underlies the encoding of (strings of) Arabic digits and subserves functions such as
the classification of Arabic numerals in terms of their cardinality. The triple-code model pro-
poses various transcoding routes between the three codes [6, 7] to shift between analogue, ver-
bal and visual magnitude representations (assessed amongst other in number naming and
writing tasks).

According to the four-step developmental model of numerical cognition [8], these three
modules develop in a quasi-hierarchical manner. Children are born with an innate nonverbal
approximate number system, representing nonsymbolic numbers along a spatially oriented
mental number line. Around preschool, children acquire the meanings of number words and
Arabic digits, by linking them to their nonverbal quantity concepts. This ultimately leads to the
establishment of a symbolic mental number line (e.g., [9, 10]). Other theories also suggest that
increasing knowledge of the count-list further assists the development of an exact symbolic
number system via order relations between symbolic numbers (see e.g., [11]). Such knowledge
of symbol-symbol associations supposedly replaces knowledge of symbol-quantity associations
as a crucial mechanism in the development of more complex mathematical skills [12].

Acquisition of the symbolic number system is highly dependent on language and other
domain-general cognitive as well as sociodemographic factors. In the present study, we will
determine the importance of spatial language for early numerical development in preschool.
To assess its unique contribution to early numeracy skills, we will account for the potentially
predictive effects of other commonly studied domain-general factors, including phonological
awareness, spatial abilities and sociodemographic factors, such as socioeconomic status (SES)
and home language. In the following, we will briefly summarise previous literature on the
importance of each of these variables for early numerical development, starting with the influ-
ences of children's sociodemographic determinants and ending with relatively recent findings
regarding the crucial role of spatial language.

## Early numerical skills and the effects of domain-general factors

Early numerical skills have been frequently shown to depend on sociodemographic factors,
such as *SES* and *home language*. Children from low SES households demonstrate weaker
numerical knowledge during the preschool years compared to their middle-income peers [13,
14]. Oftentimes, lower SES is confounded with having an immigration background, which has
also been associated with lower academic performances across many countries (e.g., [15, 16]).
Amongst others, this might be attributed to linguistic dissimilarities between native children
and those from immigrant families, especially if the home language differs considerably from
the language of instruction at school [17, 18]. Namely, second language learners were

repeatedly shown to score lower than their first language peers on linguistic as well as early numeracy and arithmetic tasks already in preschool and at the beginning of primary school [19–22]. Likewise, Bernardo and Calleja [23] reported that Filipino-English bilingual students were more likely to experience failure in solving mathematical word problems in their second compared to their first language. Such performance differences between first and second language learners can be attributed to weaker knowledge of the instruction language when it is acquired as a second language. For instance, Greisen et al. [18] showed that differences in mathematics achievement in third grade between speakers of a home language that is similar to the instruction language compared to speakers of distant home languages are fully mediated by their underachievement in reading comprehension in the instruction language (see also [17, 24]).

Apart from home language, the ability to represent, access and manipulate speech sounds within words, referred to as *phonological awareness* [25], also plays an important role in the development of young children's early (symbolic) numerical skills (see meta-analysis by [26]). Phonological awareness probably supports the acquisition and automation of number words similarly to how it facilities learning of other types of vocabulary [27]. And, according to Simmons and Singleton [28], deficits in phonological awareness should mostly impair those aspects of mathematics depending on the manipulation of verbal number codes (e.g., counting speed, number fact recall). This clearly agrees with studies repeatedly highlighting its contributing role to sequential counting skills in preschool children (e.g., [29, 30]), even when controlling for other domain-general factors such as spatial perception performances [31]. Interestingly, it was also the strongest predictor of preschool children's quantity processing skills, followed by counting and spatial perception [32]. In addition, phonological awareness in preschool was shown to predict later competencies in arithmetic word problems and written arithmetic ([33], see also [34]). It also mediated the relation between verbal working memory and number transcoding in older children attending second to fourth grade [35].

In addition to phonological awareness, s*patial abilities* have been widely recognized as being concurrently and longitudinally associated with a large variety of mathematical outcome variables across development, including early number knowledge in young children (e.g., [36–38]). Ansari et al. [39] showed that spatial visualization was more important than language ability in the development of cardinality understanding in typically developing 3-year-old children. Furthermore, Zhang et al. [40] reported that children's spatial visualization skills in preschool predicted arithmetic performances in first grade as well as later growth through third grade via their counting sequence knowledge assessed in first grade. This also agrees with the findings of Barnes et al. [41], indicating that spatial visualization skills in 5-year-olds were associated concurrently with oral counting as well as nonsymbolic arithmetic. Zhang [42] reported that spatial perception in 3-year-old children uniquely contributed to early number competence, as measured using tasks such as counting sequence, counting principles, number recognition, number comparison, nonverbal calculation, story problems, and number combinations. Moreover, Cornu et al. [43] showed that the relation between spatial perception and arithmetic in preschool children was partially mediated by their number line estimation performances. Spatial perception was also found to be an important predictor of preschool children's verbal number skills, as indexed by aggregating performances on counting and number naming tasks [31, 44]. These findings were recently extended by Liu and Zhang [45], highlighting a reciprocal relation between preschool children's spatial perception skills and counting sequence knowledge, especially for backward counting. Mix et al. [37] singled out mental rotation as an important predictor of early numerical skills in preschool children. Moreover, mental rotation skills at 5 years of age significantly predicted number line estimation performances one year later, which in turn positively affected approximate symbolic calculations at 8-years-

old [46]. Likewise, Frick [47] observed strong relations between mental rotation and arithmetic operations at the beginning of formal schooling.

## Early numerical skills and the effects of spatial language

More recent studies have also focussed on the language of spatial concepts and relationships, referred to as *spatial language* (for a recent review, see [48]), in the context of early numerical development. Spatial language is a means to verbally symbolize spatial thinking, providing a symbolic system to represent and reason about space. According to Cannon et al. [49], spatial language can be differentiated into eight categories, namely (1) spatial dimensions (i.e., size– big, small); (2) shapes (i.e., square); (3) locations and directions (to describe relative positions); (4) orientations and transformations (i.e., turn right); (5) continuous amount (i.e., whole, piece, portion, etc.); (6) deictics (i.e., here, there, where, etc.); (7) spatial features and properties (i.e., side, curve, round, line, etc.); (8) pattern (i.e., next, after, sequence, increase, decrease).

Spatial language begins to develop within the first year of life, when infants start to form basic representations and categories for spatial terms [50, 51]. Children normally produce and comprehend simple spatial prepositions by 2 years of age [52, 53] and more complex prepositions at around 5-years-old [53, 54]. The rate of this progress was shown to depend on home environment in that higher SES backgrounds were associated with better spatial language in 3-year-olds [55] as well as in preschool children [31, 56]. This relation might be explained by the observation that parents from lower SES backgrounds use fewer spatial terms when interacting with their children at home [57], which in turn was shown to predict the children's own spatial language [58].

Interestingly, languages differ quite widely with respect to how spatial language is structured. For instance, Bowerman [59] showed that containment and support relations are associated with different numbers of spatial prepositions across languages. Namely, Spanish has a single term "en", which is equivalent to the English spatial terms "in" and "on". Moreover, Dutch has one term "aan" corresponding to situations such as "a handle on a door", while "op" is used for situations such as "a book on a shelf". This structural diversity across languages could become a problem especially for second language learners, when they acquire and use spatial terms in their second language [60].

Importantly, spatial language was shown to not only positively associate with spatial abilities (e.g., [58, 61]) but to also relate to numerical skills in young children (for a recent review, see [62]). The knowledge of spatial terms thus seems to facilitate spatial cognition more broadly (for a review, see [63]) by also affecting the spatial aspects of numerical processing. For instance, mathematical language, assessed using Purpura and Logan's [64] measure of mathematical content language including six quantitative items (i.e., take away, a little bit, more, less, most, and fewest) and ten spatial items (i.e., nearest, under, first, far, below, front, middle, end, last, and before), was found to predict a variety of numeracy outcomes, over and above general language skills [56, 65]. The same measure was also recently shown to account for the relation between home numeracy environment (i.e., home activities such as counting objects) and children's mathematical abilities [66]. Mathematical language, indexed with 24 quantitative (e.g., half, equal, more) and spatial (e.g., behind, between, opposite) items from the "sentence structures" subtest of the Language Test for All Children [67], was also shown to mediate the relation between general language and numerical development in preschool [68]. Moreover, training mathematical language (focussing on both quantitative and spatial terms) via story book reading in 3- to 5-year-olds was shown to improve general mathematical knowledge compared to a business-as-usual control group ([69] but see [70]). Furthermore, when focussing more specifically on spatial language, Bower et al. [71] reported that the

comprehension of spatial relation terms (i.e., under, above, between, up, in, on, down, behind, below, middle, in front of, next to, on top of, and upside down) in 3-year-olds mediated the relation between spatial abilities and early numerical skills, especially in girls. Hawes et al. [72] also indicated significant correlations between spatial language comprehension and numeracy outcomes in 6-year-olds. In that study, children were required to identify their left vs. right hand, the location of an object in relation to a box, and various shapes and figures (e.g., cube). Moreover, Georges et al. [31] found that preschool children's spatial language skills, assessed by the production and comprehension of locative prepositions (i.e., on, left, before, in, right, behind, above, and under), predicted their verbal number skills, reflecting performances on both counting and number naming tasks, even when accounting for the influences of phonological awareness, spatial perception as well as age, gender, and socioeconomic status. The importance of spatial language, as assessed in Georges et al. [31], for verbal number skills in young children was also confirmed by Lindner et al. [73], after controlling for potentially confounding variables such as spatial perception, vocabulary knowledge, age and gender. In their study, spatial language even predicted the acquisition of verbal numerical skills 6 months later, thereby further highlighting the importance of the mastery of spatial relation terms in the development of verbal number knowledge. This also agrees with the observation that in 4- to 5-year-old children the knowledge of the spatial terms "in front", "left", "right", and "behind", when used in an egocentric reference frame, significantly related to their number word comparison performances, even when accounting for socioeconomic background, mental rotation skills, as well as the understandings of absolute magnitude and numerical sequences [74]. Turan and colleagues [75] recently extended these findings by showing that spatial language, as assessed by a measure focussing on 12 spatial items (i.e., far away, at the end, front, closest to, on, first, last, under, behind, in front of, under, middle) adapted from Purpura and Logan [64], was significantly associated with preschool children's geometry performances and also partially mediated the effects of general language on the latter. Finally, Gilligan-Lee et al. [76] observed positive relations between spatial language and standardised mathematics skills as well as estimations on 0–100 number lines but not dot comparison performances in slightly older children aged 6 to 10 years, after controlling for spatial abilities, vocabulary knowledge, age, and grade. To account for the slightly older age range of the children, they adapted the spatial language measure of Farran and Atkinson [77] to include the following 12 spatial relation terms: above, below, right, left, between, around, through, higher, lower, closer, further, parallel.

## The present study

The aim of the present study was to assess more extensively the importance of spatial language for early numerical development in preschool. Since most previous studies investigating the role of spatial language in numerical cognition have considered locative prepositions (see e.g., [31, 71, 73, 74, 76]), i.e., spatial terms within the category "locations and directions" [49], the present study also specifically focussed on the latter. This category also seems most relevant when considering that numerical magnitudes are supposedly mentally represented on a spatially oriented linear continuum, the mental number line, with small/large numbers located on the left/right, respectively [78]. Numbers can thus be spatially localized in relation to each other depending on their cardinality and/or ordinality (e.g., 2 is before, next to, on the left of 3). Since the spatial mapping of numerical magnitude representations seems like a plausible candidate through which spatial language could facilitate numerical cognition, assessing the knowledge of spatial relation terms seems most pertinent in this case (see also [73] for a similar rationale).

Considering that previous studies in preschool children [31, 73] have highlighted the crucial role of spatial language for verbal number skills, as indexed by aggregated scores on counting and number naming tasks, this study firstly aimed to explore whether spatial language significantly predicts both counting and number naming or whether the aforementioned relation between spatial language and verbal number skills was driven by only one of these variables.

In addition, we aimed to further assess whether the predictive value of spatial language extends beyond verbal number skills to numerical subdomains without explicit verbal component ([75], see [76] for findings in older children). To determine whether spatial language affects early mathematical development more broadly, we additionally assessed the children's performances on tasks involving numerosity comparisons, symbolic magnitude classifications and ordinal judgments as well as number writing. According to the findings of Gilligan-Lee et al. [76] in older children, we hypothesized that spatial language would be less relevant for numerosity comparisons, as these authors did not find a relation between spatial language and performances on a dot comparison task. They suggested that spatial language might offer an effective strategy for completing numerical tasks where verbal coding of spatial relations between (symbolic) numbers is particularly useful. As such, we expected spatial language to be especially related to counting as well as symbolic magnitude classifications and ordinal judgments, since these tasks all require children to actively keep track of the cardinal and/or ordinal relations between symbolic numbers.

To determine the unique contributions of spatial language to these numerical skills, we performed regression analyses, where we added in a first step both spatial abilities and phonological awareness as potentially confounding predictors. While the literature delineates many theoretical distinctions between different spatial subskills, the present study focused on Uttal et al.'s [79] top-down model of spatial topology. These authors adopted Newcombe and Shipley's [80] theory-driven framework, outlining a 2 x 2 classification of spatial abilities by crossing two dimensions (see [81], for a review). The static-dynamic dimension reflects the observation that spatial tasks can involve objects arranged in either stable positions (static) or motion (dynamic). The intrinsic-extrinsic dimension reflects the fact that spatial relations can be inherent to an object and its parts (intrinsic) or can exist among different objects as well as between an object and its reference frame (extrinsic). In accordance with preschool children's spatial competencies, we included measures indexing intrinsic-static, intrinsic-dynamic as well as extrinsic-static spatial categories, thereby extending previous studies that only considered the potentially predictive effect of spatial perception, an extrinsic-static spatial skill [31, 73]. This allowed us to get a better understanding of 1) the relative contributions of these different spatial abilities to early mathematical development before and after controlling for spatial language and 2) the predictive value of spatial language for early numerical skills when accounting for the multi-dimensionality of spatial ability. Based on recent findings highlighting the predictive value of spatial language also in the presence of covariates (e.g., [31, 73, 76]), we hypothesized that spatial language would still relate to at least some of the current numerical measures even after controlling for spatial abilities and phonological awareness.

Before adding spatial language in the final step of the models, we additionally controlled in a second step for sociodemographic factors, including age and SES [31] as well as the children's home language and as such indirectly their mastery of the language of instruction in school. The present study was conducted in Luxembourg, where children are thought in Luxembourgish during the preschool years, which is mandatory from age four. Yet, due to the high immigration rate in Luxembourg (47.1% foreigners in 2022) [82], only about 33% of Luxembourg's school population speaks Luxembourgish at home [83]. Since second language learners not only perform worse in math tasks compared to their native peers (e.g., [17–22]), but also likely have difficulties acquiring spatial terms in their second language [60], we grouped the present

sample based on whether children spoke the language of instruction (i.e., Luxembourgish) with their mother at home or not. We considered this variable as a potentially confounding factor when assessing the effects of spatial language on early numerical skills. To the best of our knowledge, the effects of instruction language knowledge have never been considered when assessing relations between spatial language and mathematical performances. Including this variable in our design might thus lead to some interesting observations, shaping the path for further investigations.

## Materials and methods

The study was approved by the Ethics Review Panel (ERP) of the University of Luxembourg.

### Participants

A total of 172 children participated in this study. Pupils were recruited from Cycle 1.2 (corresponding to the second year of preschool) of eight different public schools in Luxembourg. Data was collected at the end of the school year (i.e., in June and July) and accessed for research purposes immediately after data collection was finished. Parents' written informed consent was obtained prior to the start of the study, and all children gave their verbal consent to participate. Prior to data analysis, nine children were excluded since they did not respond in Luxembourgish on the spatial language production task, which is the language of instruction in preschool and also the language used throughout this study (6 children produced the spatial terms in French and 3 in English). An additional 7 children were removed due to experimenter error on the spatial language production task. Finally, one pupil was excluded since he/she had difficulties understanding task instructions in Luxembourgish due to a lack of Luxembourgish language knowledge. All analyses were thus conducted on 155 healthy preschool children, of whom 73 were boys. All data were pseudonymized with only the main author having access to the pseudonymization list.

Sociodemographic information in terms of socioeconomic status (SES) and language background was provided by a parental questionnaire handed to the parents alongside with the parental consent.

SES was measured by a parental questionnaire asking about main professional occupation and qualification/skill level through 10 response alternatives (e.g., unemployed, agricultural worker, scientist or related profession, etc.). Parents' occupational level was operationalized through the International Standard Classification of Occupation (ISCO-08) and the corresponding International Socio-Economic Index of Occupational Status (ISEI, [84], see also [85]). Responses were collected from both parents, but for further analyses a family index was computed by considering the highest value out of the two (Highest Socio-Economic Index, HISEI).

The children's language backgrounds were assessed by asking both parents which language they predominantly spoke with their child (see Table 1). Parents of 6 children did not respond to this question. 71 children spoke Luxembourgish with their mother at home, of which 52 were monolingual (i.e., also spoke Luxembourgish with their father at home). Of the remaining 78 children that were not raised in Luxembourgish by their mother, 52 had a single (non-

**Table 1. The distribution of children based on their language profiles.**

| Mother language | Monolingual | Multilingual |
|---|---|---|
| Luxembourgish | 52 | 19 |
| Non-Luxembourgish | 52 | 26 |

Luxembourgish) mother tongue, 8 spoke Luxembourgish with their father, and 18 spoke different (non-Luxembourgish) languages with their mother and father. All the children included in the analyses were able to follow task instructions in Luxembourgish.

## Procedure

Children were tested individually in a quiet room inside their school building. Tests were administered during two sessions of approximately 35 minutes each. Testing was carried out during the last term of the school year (i.e., in June and July) by trained student helpers. All children performed the tests in the same fixed order as follows–Session 1: forward counting, backward counting, number naming, number writing, numerosity comparison, figure ground, spatial relations; Session 2: symbolic magnitude classification, children's mental transformation task (CMTT), symbolic ordinal judgment, spatial language, phonological awareness. The numerosity comparison, symbolic magnitude classification and symbolic ordinal judgment tasks were administered using an ACER Spin laptop. The remaining tasks were paper-and-pencil-based. All task instructions were provided in Luxembourgish, the language of instruction in preschool.

## Measures

**Numerical skills.**   Children's numerical skills were assessed using seven numerical tasks: forward counting, backward counting, number naming, number writing, symbolic magnitude classification, symbolic ordinal judgment and numerosity comparison.

In the *forward counting task*, children had to verbally recite the number chain as high as possible. They were interrupted after their first mistake. Similar to Georges et al. ([31], see also [44]), different scores ranging from 0 to 10 were attributed depending on the last number correctly produced. If children could not count up to 5, the attributed score was 0. Counting up to a number between 5 and 9 was awarded with 1 point. Counting up to a number between 10 and 14 was awarded with 2 points. This score allocation pattern continued with always 1 additional point being awarded for every 5 numbers correctly added to the counting sequence. The maximum score that children could reach on this task was 10 points, corresponding to a counting sequence of 50 or above. One child was excluded from this experiment, as he/she refused to recite any numbers on this task.

In the *backward counting task*, children were asked to count backward from the starting numbers 4, 8, 12, and 20, always presented in that order. 1 point was awarded for correctly counting four numbers backward. Testing was discontinued after failure to count four numbers backward on two successive trials. The remaining trials were then scored as 0. The maximum score that children could reach on this task was thus 4.

In the *number naming task*, children were instructed to name Arabic numbers that were printed on a DIN A4 sheet, with one number per sheet. Target numbers were 7, 13, 5, 2, and 10, always presented in that order. Correct answers were awarded 1 point and the maximum score that could be reached on this task was 5.

In the *number writing task*, children were instructed to write down verbally presented numbers. Target numbers were 4, 6, 3, 9, and 8, always presented in that order. One point was awarded for every correctly written number with the maximum score that could thus be reached on this task being 5.

In the *symbolic magnitude classification task*, children had to judge whether a centrally presented Arabic digit was smaller or larger than 5. The design of this task was adapted from Hoffmann et al. ([78], see also [86]).

Each trial started with an empty, black-bordered square on a white background. After 1000 ms, one of eight possible stimuli (Arabic digits 1, 2, 3, 4, 6, 7, 8, and 9), presented in black in font Arial point size 48, appeared in the center of the square and remained until response or until 5000 ms had elapsed. The stimuli were presented in a pseudo-random order, no number appeared twice in a row, and the correct response could be on the same side no more than two times consecutively. Each child completed two blocks. First, they had to press the "A" or "L" key of a standard QWERTZ keyboard for numbers smaller or larger than 5, respectively. This stimulus-response mapping was then reversed in the second block. Response keys were highlighted with colored stickers and a cartoon mouse or elephant tag was placed above the answer key to indicate "smaller" or "larger", respectively. Each block contained 40 trials with 5 repetitions per digit. The inter-stimulus interval was a blank screen of 1000 ms. Every block started with 8 to 16 training trials, depending on response accuracy. An accuracy threshold of 80% correct answers needed to be reached in order to proceed directly to the experimental trials after 8 training trials; if the threshold was not reached, another 8 training trials were administered before the experimental trials started. Feedback was given only during the practice sessions by means of happy and sad smiley faces after each round of 8 practice trials.

Children's performances on this task were analyzed in terms of accuracy (number of correct items, maximum = 80). RTs < 200 ms were considered as anticipations and discarded as errors (average = 0.32 trials, SD = 1.10, 0.46% of all correct trials). One child was excluded from this experiment, because the task ended prematurely due to a technical error.

In the *symbolic ordinal judgment task*, children had to judge whether three horizontally displayed Arabic digits were presented in correct order (numerically increasing from left-to-right) or not.

Each trial started with the central presentation of a white fixation cross for 300 ms on a grey background. This was immediately followed by the horizontal display of three Arabic digits. Numerical distances between the digits were fixed to one. On half of the trials, the three digits were all in numerically increasing order (from left-to-right), while on the remaining trials, digits were either in decreasing (from right-to-left) or mixed order. Trials were randomly presented. Children were instructed to press the "A" or "L" key of a standard QWERTZ keyboard if the three digits were all in numerically increasing order or not (i.e., decreasing, or mixed order), respectively. A green or red sticker was placed above the "A" or "L" key to indicate "correct" or "incorrect", respectively. Stimuli remained on the screen until the child responded. The experiment comprised 12 trials (i.e., 6 in increasing order, 3 in decreasing order, and 3 in mixed order). The inter-stimulus interval consisted of a blank screen for 700 ms. Each child completed the same three practice trials prior to the start of the experiment. Practice trials included two correct (increasing) sequences and one incorrect (decreasing) sequence. Practice was repeated once if two out of three trials were incorrect. Feedback was given only on the practice trials by means of a happy or sad smiley face after each correct or incorrect response, respectively.

Children's performances on this task were analyzed in terms of accuracy (number of correct items, maximum = 12). There were no anticipations in terms of RTs < 200 ms on this task, such that all correct trials were included in the analyses. 5 children were excluded from this experiment, because no fixation cross was shown due to experimental error. Nonetheless, conclusions remained identical when including these participants.

Children performed significantly better on random order trials than on ascending or descending order trials. In other terms, children had greater difficulties judging a correct increasing order as correct and an incorrect decreasing order as incorrect than an incorrect random order as incorrect (F(1.79, 266.16) = 4.59, p = .014, $\eta p2$ = .03, normed accuracies: ascending = .78, descending = .76, random = .83). Accuracies across the different orders were significantly related.

In the *numerosity comparison task*, children had to determine the largest number of dots out of two arrays displayed simultaneously on their left and right side.

Each trial started with the central presentation of a black fixation cross for 500 ms on a grey-ish blue (RGB 188, 185, 255) background. This was immediately followed by the display of two dot arrays on the left and right side of the screen. The arrays only remained for a maximal duration of 800 ms to prevent participants from counting the dots. An active mask was used until response to suppress retinal persistence. Responses were given by pressing the "A" or "L" key of a standard QWERTZ keyboard if the correct response was on the left or right side, respectively. The experiment comprised 60 trials that were randomly presented. The inter-stimulus interval consisted of a blank screen for 400 ms. Each child completed the same four practice trials prior to the start of the experiment. Feedback was given only on the practice tri-als by means of the word "correct" presented in green or the word "incorrect" presented in red after each correct or incorrect response, respectively.

The 60 stimulus pairs were equally but randomly drawn from one of four numerical ratio bins: 1:2 (4 vs. 8, 5 vs. 10, 6 vs. 12, 7 vs. 14 dots), 2:3 (4 vs. 6, 6 vs. 9, 8 vs. 12, 10 vs. 15 dots), 3:4 (6 vs. 8, 9 vs. 12 dots), 6:7 (6 vs. 7, 12 vs. 14 dots). The four practice trials included one pair from each ratio bin. The number of dots per array varied between 4 and 14 and the total num-ber of dots on the screen ranged between 10 (i.e., 4 vs. 6 dots) and 26 (i.e., 12 vs. 14 dots). The position of the more numerous dot array of the pair (i.e., the correct response) was randomly assigned to the left or to the right throughout the experiment.

Dot arrays were generated with the NASCO application [87]. Stimulus pairs were divided into four categories. In a first category, total area (TA) and convex hull (CH) were equalized across the stimulus pair; in a second category, TA, and mean occupancy (MO) were equalized; in a third category, item size (IS) and CH were equalized; and in a fourth category, IS and MO were equalized. All the dots were the same size within an array. Across the stimulus set, mean IS was 523.56 px, Range (R) [380–804 px]; mean TA was 4541.50 px, R [2124–7965 px]; mean CH was 85805 px, R [52000–154000 px]; and mean MO was 10537 px, R [5714–18000 px].

Children's performances on this task were analyzed in terms of accuracy (number of correct items, maximum = 60). RTs < 200 ms were considered as anticipations and discarded as errors (average = 0.58 trials, SD = 1.67, 1.32% of all correct trials). Two children were excluded from this experiment, because the task either ended prematurely or was aborted by the child halfway through. A repeated measures ANOVA revealed that there was a main effect of ratio ($F_{(3, 456)}$ = 51.25, p < .001, $\eta p2$ = .25), with better performances on larger ratio trials (R1.17: $\bar{x}$ = 9.47 trials, R1.33: $\bar{x}$ = 10.42 trials, R1.50: $\bar{x}$ = 11.36 trials, R2.00: $\bar{x}$ = 11.76 trials). Performances were significantly correlated across ratios. We did not compute the Weber fraction, as recent evi-dence suggests that it is not more informative than accuracy [88].

**Spatial abilities.** Children's spatial abilities were assessed using three different spatial tasks: the figure ground task, a shortened version of the children's mental transformation task (CMTT), and the spatial relations task. The figure ground and spatial relations tasks were taken from the paper-and-pencil-based FEW-2 ("Frostigs Entwicklungstest der visuellen Wahrnehmung 2") [89], which is a standardized German developmental test battery equiva-lent to the DTVP-2 (Developmental Test of Visual Perception 2). According to the 2x2 taxon-omy of Newcombe and Shipley [80], the figure ground and spatial relations tasks can be used to index intrinsic-static and extrinsic-static spatial abilities, respectively. The FEW-2 does not include tasks assessing intrinsic-dynamic spatial abilities. For this reason, we included a short-ened version of the CMTT. In the original task [90], half of the items required mental rotation, while the remaining half focused on mental translation. Since we used this task as an index of intrinsic-dynamic spatial abilities (i.e., mental rotation), we opted to include only those items involving mental rotation (see also [91, 92]).

In the *figure ground task*, children were presented with a 2-D target figure consisting of multiple superimposed forms. They had to identify the individual forms constituting the target figure from 5–10 alternatives depicted below the target figure. The target figure either included only the forms that had to be identified or additionally comprised a distracting background image. The task consisted of 18 trials and one point was awarded for each trial on which all the individual forms that were hidden within the target figure were correctly identified (maximum score = 18). The task was preceded by three practice trials, where feedback was given. Cronbach's α was .61 and the Spearman-Brown corrected split-half reliability estimate based on the odd-even method was .66.

In the shortened version of the *CMTT*, children were presented with a target shape divided into two halves that were displayed above a 2×2 array of four choice shapes. Children were asked to identify the shape out of the four choice shapes that would result from piecing together the two halves of the target shape. The shortened task consisted of 16 mental rotation trials and one point was awarded for every correct answer (maximum score = 16). No feedback was given on any trial. Cronbach's α was .65 and the Spearman-Brown corrected split-half reliability estimate based on the odd-even method was .69 (see also [91]).

In the *spatial relations task*, children were presented with 10 grids of regularly arranged dots. The size of the grid varied between 2x2 to 4x6 dots. Whitin each grid, some of the dots were connected by lines. The number of dots connected by lines varied between 2 and 10. The children were asked to reproduce the lines connecting the dots in an empty grid displayed immediately below the grid they had to copy using a pencil. Children were awarded one point for every correctly re-produced line (i.e., for every line correctly connecting the two dots). The maximum score that could be reached on this task was 34 (i.e., they had to reproduce a maximum of 34 lines divided among 10 grids). The task was preceded by three practice trials where only two dots had to be connected on every trial. Feedback was given only on practice trials. Cronbach's α was .65 and the Spearman-Brown corrected split-half reliability estimate based on the odd-even method was .76.

**Phonological awareness.**   Children's phonological awareness was assessed using a task consisting of four subtests [31]: phoneme detection part 1 and 2, phoneme blending, and rhyme identification. For every subtest, correct and incorrect answers were awarded 1 and 0 points, respectively. On the first part of the *phoneme detection task*, children had to indicate whether nouns started with the initial sound /a/. On the second part, children needed to judge whether nouns contained the sound /i/. On each part, a training item preceded four test trials (two "yes" and two "no" trials). On the *phoneme blending task*, children were presented with the phonemes of a word and had to pronounce the word obtained by joining the phonemes. One training item preceded three test items. Finally, on the *rhyme identification task*, children had to indicate whether two presented non-words rhymed or not (e.g., "tuk–nuk" vs. "mau–méi"). Two training items preceded eight test items (four rhymes and four non- rhymes).

To index phonological awareness, a single score was calculated by adding the children's scores (normed to 1) across the four subtests. Cronbach's α was .69. The internal structure was confirmed by an exploratory principal component analysis yielding a one-factor solution describing the performances in all four subtests. The scores across the different subtests were also significantly correlated.

**Spatial language.**   As in Georges et al. [31], children's spatial language was assessed using spatial language production (based on [77], see also [93]) and comprehension (see [76]) tasks. For both tasks, correct and incorrect answers were coded as 1 and 0, respectively. In the *production task*, a teddy bear was placed in different positions relative to a box. The children were then asked to name the spatial term describing the teddy bear's location with respect to the box. The task consisted of 8 trials assessing 8 different spatial locations in the following order: on, left, before, in, right, behind, above and under. The same spatial terms were assessed in the

**Table 2. Percentage of children yielding correct responses on each of the trials on the spatial language production and comprehension tasks.**

| Spatial term | Production | Comprehension |
|---|---|---|
| On | 65 | 97 |
| Left | 23 | 94 |
| Before | 55 | 92 |
| In | 87 | 97 |
| Right | 20 | 80 |
| Behind | 66 | 87 |
| Above | 17 | 63 |
| Under | 63 | 98 |

*comprehension task* in the following order: under, before, left, behind, above, on, right, in. On each of these 8 trials, four pictures arranged on a DIN A4 sheet depicted different relations between a "target" object (e.g., bird, apple, book) and a "reference" object (e.g., tree, table, box). On every trial, children had to point towards the picture depicting the respective afore-mentioned spatial relations between the objects. The comprehension task was preceded by one practice trial assessing the spatial term "behind". The production task was always administered prior to the comprehension task to avoid the priming of the spatial terms.

The children's performances for each spatial term on the spatial language production and comprehension tasks can be found in Table 2.

On the production task, only 23% and 20% of the children correctly responded on the trials assessing the spatial terms "left" and "right", respectively, with 17% of the children getting both items correct. This is similar to our previous findings [31] and in line with the fact that young children still have difficulties using these labels appropriately until 7–11 years of age ([94], see also [95]). Here, it should be noted that children who already correctly produced both left and right (i.e., n = 26, 17%) did not outperform their peers on the remaining six items of the spatial language production task and they showed similar performances than their peers on the spatial language comprehension task. Moreover, they did not differ in any of the other cognitive or sociodemographic variables included in this study. There were also no significant associations between the ability to produce left and right and the language the children spoke with their mother at home (out of the 26 children that could produce both left and right, 13 spoke Luxembourgish with their mother at home).

As in Georges et al. [31], we therefore decided to code the children's responses less strictly on the "left" and "right" items of the production task, also awarding 1 point when the child said "next to" or "on the side of" the box instead of "left" or "right". In that case, 74% and 75% of the children responded correctly when tasked with "left" and "right", respectively. Reliability was significantly higher across all items of the production task when coding "left" and "right" less strictly (Cronbach's α = .73) compared to when only considering "left" and "right" responses as correct (Cronbach's α = .57). Children did not encounter any major difficulties with the "left" and "right" items on the comprehension task (see Table 2).

More than half of the children yielded correct responses on each of the remaining trials except for the trial assessing the spatial term "above" in the production task, which was correctly responded to only by 17% of the children (see Table 2). To be in accordance with Georges et al. [31], we decided to keep this item when computing the children's spatial language scores. Removing it from the spatial language score did not change any of the main conclusions of the present study. Reliability on the spatial language production task did not change when excluding the item assessing the spatial term "above" (Cronbach's α = .73).

The average total score (summed across all 8 items) on the spatial language production task (when coded less strictly for the items left and right) was significantly lower than the average total score on the comprehension task ($F(1, 154) = 211.50$, $p < .001$, $\eta_p^2 = .58$, production = 5.00, comprehension = 7.08). This suggests that despite the children's ability to understand most of the spatial terms, they were not yet able to produce them. Scores on the production and comprehension tasks were significantly correlated ($r = .53$, $p < .001$).

For subsequent data analyses, a total score indexing spatial language was calculated by summing the scores of the production and comprehension tasks. Cronbach's α across all items of both tasks was .75.

**Covariates.** Age, SES, as indexed by HISEI, as well as language spoken with mother, in terms of Luxembourgish or not (coded as 1 or 0, respectively), were included as covariates in the present study. It should be noted that results were similar when grouping children based on whether or not they spoke Luxembourgish with either their mother or father at home. Age was calculated by averaging the children's ages across both testing sessions. Information about age, HISEI and language background was missing for one, seven and six children, respectively. These children were excluded from all analyses including one or more of those covariates.

Gender was not included as a covariate in the present study, since preliminary analyses with one-way repeated measures ANOVAs did not indicate any significant differences between boys and girls on any of the variables included in this study.

## Analyses

First, analyses of variance (ANOVA) and correlation analyses were conducted to assess relations between all cognitive and sociodemographic variables included in the present study. We then conducted hierarchical multiple linear regression analyses on each of the numerical skills considered in this study. Intrinsic and extrinsic spatial abilities, as indexed by performances on the figure ground, CMTT and spatial relations tasks, as well as phonological awareness were included in step 1. This allowed us to determine their relative contributions to early numerical skills prior to controlling for sociodemographic factors. The covariates age, HISEI, and language spoken with mother were then added in step 2 whenever they significantly related to the numerical outcome variable as shown by the ANOVA or correlation analyses. In step 3, spatial language was added to the regression models to investigate its unique contribution to the variance in numerical skills when controlling for spatial abilities, phonological awareness and the sociodemographic factors. The inclusion of spatial language in the final step also allowed us to ascertain how the predictive effects of the remaining cognitive and sociodemographic variables on numerical skills changed following its addition to the regression models. Multicollinearity was assessed by examining the variance inflation factor (VIF). VIF was always below 2, thus indicating no serious problems of multicollinearity. All analyses were conducted using SPSS statistical program for Windows, version 27 (SPSS Inc., Chicago, IL, USA).

## Results

Descriptive information regarding age, HISEI and performances on all cognitive tasks can be found in Table 3.

### Analysis of variance and correlation analyses

One-way ANOVAs indicated significant effects of language spoken with mother on forward and backward counting as well as on ordinal judgments. Children speaking Luxembourgish with their mother at home outperformed their peers on all three numerical tasks (forward

**Table 3. Descriptives.**

| Variable | n | Mean | SD | Theoretical range | Empirical range |
|---|---|---|---|---|---|
| Age (in years) | 154 | 6.36 | 0.34 | - | 5.57–7.71 |
| HISEI | 148 | 51.89 | 16.24 | 20.82–69.90 | 20.82–69.90 |
| Forward counting | 154 | 5.41 | 2.69 | 0–10 | 0–10 |
| Backward counting | 155 | 2.16 | 1.25 | 0–4 | 0–4 |
| Number naming | 155 | 4.56 | 0.72 | 0–5 | 2–5 |
| Number writing | 155 | 3.28 | 1.16 | 0–5 | 0–5 |
| Symbolic magnitude classification | 154 | 70.24 | 12.25 | 0–80 | 23–80 |
| Symbolic ordinal judgment | 150 | 9.45 | 2.24 | 0–12 | 3–12 |
| Numerosity comparison | 153 | 43.02 | 9.37 | 0–60 | 14–57 |
| Figure ground | 155 | 12.34 | 2.46 | 0–18 | 6–17 |
| CMTT | 155 | 10.32 | 2.99 | 0–16 | 2–16 |
| Spatial relations | 155 | 25.23 | 6.52 | 0–34 | 7–34 |
| Phonological awareness | 155 | 3.10 | 0.60 | 0–4 | 1.46–4 |
| Spatial language | 155 | 12.08 | 2.80 | 0–16 | 4–16 |

counting: $F_{(1, 146)} = 12.12$, $p = .001$, $\eta p2 = .08$, Lux $\bar{x} = 6.24$, non-Lux $\bar{x} = 4.75$; backward counting: $F_{(1, 147)} = 10.69$, $p = .001$, $\eta p2 = .07$, Lux $\bar{x} = 2.49$, non-Lux $\bar{x} = 1.85$; ordinal judgments: $F_{(1, 142)} = 5.24$, $p = .02$, $\eta p2 = .04$, Lux $\bar{x} = 9.88$, non-Lux $\bar{x} = 9.03$). Children raised by Luxembourgish-speaking mothers also performed significantly better on the phonological awareness task ($F_{(1, 147)} = 10.95$, $p = .001$, $\eta p2 = .07$, Lux $\bar{x} = 3.27$, non-Lux $\bar{x} = 2.96$) and had better spatial language than their peers not speaking Luxembourgish with their mother at home ($F_{(1, 147)} = 58.65$, $p < .001$, $\eta p2 = .29$, Lux $\bar{x} = 13.69$, non-Lux $\bar{x} = 10.72$). The former children also featured higher SES than the latter ($F_{(1, 147)} = 7.92$, $p = .006$, $\eta p2 = .05$, Lux $\bar{x} = 55.93$, non-Lux $\bar{x} = 48.62$).

Correlations between the cognitive variables as well as age and HISEI are depicted in Table 4. Spatial language significantly positively correlated with all the cognitive variables except for performances on the number writing task and on the figure ground spatial task. Numerical performances were significantly interrelated apart from numerosity comparisons, which did not relate to number naming and ordinal judgments. Performances on numerical tasks were also significantly positively associated with spatial abilities except for forward counting, which did not correlate with performances on the figure ground spatial task. All spatial abilities were significantly interrelated. Phonological awareness was significantly positively associated with all numerical and spatial variables except for figure ground spatial performances.

When considering the covariates, age significantly positively correlated only with accuracies on the magnitude classification and numerosity comparison tasks. It also positively related to performances on the figure ground and CMTT spatial tasks. Higher HISEI was associated with significantly better forward counting as well as magnitude and ordinal judgments. It also correlated with higher abilities on the CMTT spatial task, better spatial language, and phonological awareness.

## Hierarchical multiple linear regression analyses

The results of the hierarchical multiple linear regression analyses on numerical skills are depicted in Tables 5 and 6. In step 1, phonological awareness was a significant predictor of numerical skills except for number writing. Spatial relations significantly predicted backward counting as well as number writing and numerosity comparisons, while performances on the

**Table 4. Correlation analyses.**

| | 1. | 2. | 3. | 4. | 5. | 6. | 7. | 8. | 9. | 10. | 11. | 12. | 13. | 14. |
|---|---|---|---|---|---|---|---|---|---|---|---|---|---|---|
| **1. Forward counting** | 1 | .63*** | .34*** | .31*** | .16* | .26** | .27*** | .14 | .24** | .30*** | .34*** | .41*** | .04 | .22** |
| **2. Backward counting** | | 1 | .49*** | .36*** | .27** | .42*** | .36*** | .21** | .30*** | .40*** | .46*** | .47*** | .06 | .15 |
| **3. Number naming** | | | 1 | .35*** | .23** | .29*** | .15 | .18* | .27** | .18* | .37*** | .26** | .13 | .02 |
| **4. Number writing** | | | | 1 | .27*** | .19* | .22** | .25** | .18* | .39*** | .24** | .15 | .03 | .08 |
| **5. Symbolic magnitude classification** | | | | | 1 | .31*** | .30*** | .32*** | .33*** | .25** | .33*** | .29*** | .16* | .29* |
| **6. Symbolic ordinal judgment** | | | | | | 1 | .15 | .24** | .28*** | .29*** | .32*** | .34*** | -.05 | .22* |
| **7. Numerosity comparison** | | | | | | | 1 | .20* | .29*** | .41*** | .37*** | .17* | .17* | .11 |
| **8. Figure ground** | | | | | | | | 1 | .31*** | .39*** | .15 | .05 | .19* | .04 |
| **9. CMTT** | | | | | | | | | 1 | .40*** | .41*** | .16* | .19* | .20* |
| **10. Spatial relations** | | | | | | | | | | 1 | .36*** | .19* | .08 | .12 |
| **11. Phonological awareness** | | | | | | | | | | | 1 | .55*** | .11 | .22* |
| **12. Spatial language** | | | | | | | | | | | | 1 | -.14 | .27** |
| **13. Age** | | | | | | | | | | | | | 1 | -.16* |
| **14. HISEI** | | | | | | | | | | | | | | 1 |

*Note.* *p < .05

**p < .01

***p < .001.

figure ground task significantly positively related to magnitude classifications. CMTT performances did not predict any of the numerical skills when controlling for phonological awareness and the remaining spatial abilities. None of the spatial measures were related to forward counting, number naming or ordinal judgments.

In step 2, language spoken with mother was a significant predictor of forward and backward counting even when controlling for spatial abilities and phonological awareness. It did, however, no longer relate to ordinal judgements, where phonological awareness remained the only significant predictor. Conversely, phonological awareness was no longer significantly associated with forward counting after adding language spoken with mother in step 2. Instead, spatial relations emerged as a significant predictor of forward counting alongside language spoken with mother. Phonological awareness remained, however, a significant predictor of backward counting together with spatial relations after adding language spoken with mother to the model. SES was still significantly related to magnitude judgements even when controlling for spatial abilities and phonological awareness. It did, however, no longer associate with forward counting or ordinal judgments in the presence of spatial abilities, phonological awareness, and language spoken with mother. Age was no longer related to numerosity comparisons after controlling for the remaining predictors in step 2.

Finally in step 3, spatial language significantly predicted both forward and backward counting alongside spatial relations for both measures as well as phonological awareness for backward counting. Language spoken with mother was no longer a significant predictor of forward and backward counting after adding spatial language to the model. Conversely, number naming, which together with forward and backward counting was used to index verbal number skills in previous studies, was not related to spatial language when controlling for spatial abilities and phonological awareness. In the present study, it was solely predicted by the latter variable. Spatial language did also not relate to number writing, which was only predicted by spatial relations. Similarly, spatial language did not relate to numerosity comparisons, which were associated with both spatial relations and phonological awareness. Spatial language did,

**Table 5. Regression analyses on verbal numerical skills.**

| | Forward counting | | | | Backward counting | | | | Number naming | | | |
|---|---|---|---|---|---|---|---|---|---|---|---|---|
| **Model 1** | $R^2$ | F | | | $R^2$ | F | | | $R^2$ | F | | |
| | 0.15 | 6.24*** | | | 0.29 | 14.63*** | | | 0.16 | 7.14*** | | |
| **Predictors** | b | β | p | | b | β | p | | b | β | p | |
| Figure ground | 0.04 | 0.03 | 0.7 | | 0.03 | 0.06 | 0.48 | | 0.03 | 0.1 | 0.21 | |
| CMTT | 0.05 | 0.05 | 0.56 | | 0.002 | 0.004 | 0.97 | | 0.03 | 0.12 | 0.17 | |
| Spatial relations | 0.07 | 0.17 | 0.07 | | 0.04 | 0.22 | 0.008 | | -0.003 | -0.03 | 0.78 | |
| Phonological awareness | 1.14 | 0.25 | 0.005 | | 0.82 | 0.39 | <0.001 | | 0.37 | 0.31 | <0.001 | |
| **Model 2** | $R^2$ | F | $\Delta R^2$ | $\Delta F$ | $R^2$ | F | $\Delta R^2$ | $\Delta F$ | $R^2$ | F | $\Delta R^2$ | $\Delta F$ |
| | 0.21 | 6.22*** | 0.06 | 5.39** | 0.32 | 13.57*** | 0.03 | 6.94** | 0.16 | 7.14*** | - | - |
| **Predictors** | b | β | p | | b | β | p | | b | β | p | |
| Figure ground | 0.04 | 0.04 | 0.66 | | 0.03 | 0.06 | 0.46 | | 0.03 | 0.1 | 0.21 | |
| CMTT | 0.06 | 0.06 | 0.49 | | 0.01 | 0.03 | 0.74 | | 0.03 | 0.12 | 0.17 | |
| Spatial relations | 0.08 | 0.19 | 0.03 | | 0.05 | 0.25 | 0.003 | | -0.003 | -0.03 | 0.78 | |
| Phonological awareness | 0.72 | 0.16 | 0.08 | | 0.68 | 0.32 | <0.001 | | 0.37 | 0.31 | <0.001 | |
| Age | - | - | - | | - | - | - | | - | - | - | |
| HISEI | 0.02 | 0.1 | 0.22 | | - | - | - | | - | - | - | |
| Mother language | 1.19 | 0.22 | 0.008 | | 0.48 | 0.19 | 0.009 | | - | - | - | |
| **Model 3** | $R^2$ | F | $\Delta R^2$ | $\Delta F$ | $R^2$ | F | $\Delta R^2$ | $\Delta F$ | $R^2$ | F | $\Delta R^2$ | $\Delta F$ |
| | 0.24 | 6.14*** | 0.03 | 4.68* | 0.37 | 13.81*** | 0.05 | 10.49** | 0.17 | 5.98*** | 0.01 | 1.3 |
| **Predictors** | b | β | p | | b | β | p | | b | β | p | |
| Figure ground | 0.04 | 0.04 | 0.66 | | 0.03 | 0.06 | 0.46 | | 0.03 | 0.11 | 0.2 | |
| CMTT | 0.06 | 0.07 | 0.43 | | 0.01 | 0.04 | 0.66 | | 0.03 | 0.13 | 0.15 | |
| Spatial relations | 0.08 | 0.18 | 0.045 | | 0.05 | 0.23 | 0.004 | | -0.003 | -0.03 | 0.76 | |
| Phonological awareness | 0.31 | 0.07 | 0.49 | | 0.42 | 0.2 | 0.02 | | 0.3 | 0.25 | 0.01 | |
| Age | - | - | - | | - | - | - | | - | - | - | |
| HISEI | 0.01 | 0.08 | 0.32 | | - | - | - | | - | - | - | |
| Mother language | 0.72 | 0.13 | 0.14 | | 0.17 | 0.07 | 0.41 | | - | - | - | |
| Spatial language | 0.21 | 0.22 | 0.03 | | 0.13 | 0.29 | 0.001 | | 0.03 | 0.1 | 0.26 | |

*Note.* *p < .05

**p < .01

***p < .001.

however, positively affect magnitude classifications alongside performances on the figure ground task as well as age and SES. Phonological awareness was no longer a significant predictor of magnitude classifications when controlling for spatial language. Similarly, phonological awareness no longer predicted ordinal judgments after the inclusion of spatial language in step 3, which tended to be the only significant predictor.

## Discussion

### The relation of spatial language to early numerical skills

Previous studies in preschool children have highlighted the importance of the knowledge of locative prepositions for verbal number skills, as indexed by aggregating scores on counting and number naming tasks [31, 73]. The present study firstly aimed to determine whether spatial language significantly relates to both measures when indexed separately. Analyses show that spatial language significantly predicted both forward and backward counting when

**Table 6. Regression analyses on numerical skills without explicit verbal component.**

| | Number writing | | | | Symbolic magnitude classification | | | | Symbolic ordinal judgment | | | | Numerosity comparison | | | |
|---|---|---|---|---|---|---|---|---|---|---|---|---|---|---|---|---|
| **Model 1** | R² | F | | | R² | F | | | R² | F | | | R² | F | | |
| | 0.18 | 8.08*** | | | 0.19 | 8.57*** | | | 0.17 | 6.92*** | | | 0.24 | 11.29*** | | |
| **Predictors** | b | β | p | | b | β | p | | b | β | p | | b | β | p | |
| Figure ground | 0.06 | 0.12 | 0.15 | | 1.19 | 0.24 | 0.005 | | 0.12 | 0.13 | 0.12 | | 0.18 | 0.05 | 0.54 | |
| CMTT | -0.02 | -0.04 | 0.67 | | 0.57 | 0.14 | 0.12 | | 0.05 | 0.07 | 0.45 | | 0.3 | 0.1 | 0.26 | |
| Spatial relations | 0.06 | 0.32 | <0.001 | | -0.003 | -0.002 | 0.99 | | 0.04 | 0.1 | 0.27 | | 0.41 | 0.29 | 0.001 | |
| Phonological awareness | 0.24 | 0.13 | 0.13 | | 4.61 | 0.23 | 0.008 | | 0.98 | 0.26 | 0.003 | | 3.13 | 0.2 | 0.01 | |
| **Model 2** | R² | F | ΔR² | ΔF | R² | F | ΔR² | ΔF | R² | F | ΔR² | ΔF | R² | F | ΔR² | ΔF |
| | 0.18 | 8.08*** | - | - | 0.25 | 7.89*** | 0.06 | 5.46** | 0.21 | 5.82*** | 0.04 | 3.2* | 0.24 | 9.3*** | 0.01 | 1.29 |
| **Predictors** | b | β | p | | b | β | p | | b | β | p | | b | β | p | |
| Figure ground | 0.06 | 0.12 | 0.15 | | 1.13 | 0.23 | 0.007 | | 0.12 | 0.14 | 0.12 | | 0.13 | 0.03 | 0.67 | |
| CMTT | -0.02 | -0.04 | 0.67 | | 0.35 | 0.09 | 0.33 | | 0.05 | 0.06 | 0.5 | | 0.26 | 0.08 | 0.33 | |
| Spatial relations | 0.06 | 0.32 | <0.001 | | 0.001 | 0.001 | 0.99 | | 0.04 | 0.11 | 0.21 | | 0.41 | 0.29 | 0.001 | |
| Phonological awareness | 0.24 | 0.13 | 0.13 | | 3.71 | 0.19 | 0.03 | | 0.73 | 0.2 | 0.04 | | 3.05 | 0.2 | 0.02 | |
| Age | - | - | - | | 5.03 | 0.13 | 0.09 | | - | - | - | | 2.4 | 0.09 | 0.26 | |
| HISEI | - | - | - | | 0.18 | 0.24 | 0.002 | | 0.02 | 0.13 | 0.11 | | - | - | - | |
| Mother language | - | - | - | | - | - | - | | 0.57 | 0.13 | 0.13 | | - | - | - | |
| **Model 3** | R² | F | ΔR² | ΔF | R² | F | ΔR² | ΔF | R² | F | ΔR² | ΔF | R² | F | ΔR² | ΔF |
| | 0.18 | 6.45*** | 0.001 | 0.11 | 0.28 | 7.58*** | 0.02 | 4.51* | 0.23 | 5.61*** | 0.02 | 3.61# | 0.24 | 7.71*** | 0 | 0.06 |
| **Predictors** | b | β | p | | b | β | p | | b | β | p | | b | β | p | |
| Figure ground | 0.06 | 0.12 | 0.15 | | 1.1 | 0.22 | 0.008 | | 0.12 | 0.13 | 0.12 | | 0.13 | 0.03 | 0.67 | |
| CMTT | -0.01 | -0.04 | 0.69 | | 0.42 | 0.1 | 0.25 | | 0.05 | 0.07 | 0.45 | | 0.25 | 0.08 | 0.34 | |
| Spatial relations | 0.06 | 0.32 | <0.001 | | 0 | 0 | 0.99 | | 0.04 | 0.11 | 0.24 | | 0.41 | 0.29 | 0.001 | |
| Phonological awareness | 0.21 | 0.11 | 0.27 | | 1.54 | 0.08 | 0.43 | | 0.41 | 0.11 | 0.28 | | 3.26 | 0.21 | 0.03 | |
| Age | - | - | - | | 5.96 | 0.16 | 0.04 | | - | - | - | | 2.32 | 0.08 | 0.28 | |
| HISEI | - | - | - | | 0.16 | 0.22 | 0.006 | | 0.02 | 0.11 | 0.16 | | - | - | - | |
| Mother language | - | - | - | | - | - | - | | 0.25 | 0.06 | 0.54 | | - | - | - | |
| Spatial language | 0.01 | 0.03 | 0.74 | | 0.83 | 0.19 | 0.035 | | 0.16 | 0.2 | 0.06 | | -0.08 | -0.02 | 0.8 | |

*Note.* *p < .05
**p < .01
***p < .001.

controlling for phonological awareness, spatial abilities and sociodemographic factors, but did not relate to number naming, which was significantly predicted only by phonological awareness. These findings thus suggest that the previously reported relation between verbal number skills and spatial language was mainly driven by the positive effect of the latter variable on counting skills. Spatial language thus supports the production of number words only in the context of reciting the number chain (forwardly and backwardly), but not when transcoding from the visual Arabic number form to the auditory-verbal word frame.

Secondly, we aimed to determine whether the predictive value of spatial language extends beyond verbal number skills to other numerical subdomains in preschool children, and as such affects pre-mathematical development more broadly. Results show that spatial language significantly predicted the ability to classify the magnitude of symbolic numbers with respect to that of a fixed referent beyond the effects of phonological awareness, spatial abilities and sociodemographic factors. This extends the recent findings of Lindner et al. [74] to the visual domain, as these authors showed that the use of egocentric reference frames in spatial language

predicted magnitude comparisons of symbolic number words. Spatial language also tended to be the only significant predictor of symbolic ordinal judgments. These findings, however, seem to disagree with recent observations, indicating that spatial language did not relate to preschool children's knowledge of numerical sequences [74]. In that study, spatial language did not correlate with the ability to either state predecessors and successors of given number words or identify missing dice patterns of a series. The latter finding involving nonsymbolic dice pattern seems, however, in line with the present outcomes, showing that spatial language did not contribute to numerosity comparisons. The absence of a predictive effect of spatial language when comparing nonsymbolic number arrays also agrees with the study of Gilligan-Lee et al. [76], indicating that in older children aged 6 to 10 years spatial language was not predictive of performances in a dot comparison task when also controlling for spatial and verbal abilities. Finally, similar to the aforementioned results concerning number naming, spatial language did not relate to number writing, requiring the transcoding from number words to written Arabic digits.

Overall, the present findings thus highlight the importance of spatial language for early numerical abilities beyond verbal number skills and suggest a more general role in (symbolic) numerical development in preschool. Although a causal direction cannot be determined form the present results, from a theoretical perspective, we can propose three mechanistic explanations for the predictive effect of spatial language based on its differential role depending on the numerical outcome variable.

Firstly, recent studies assessing the importance of spatial language for numerical skills have considered the *spatial mapping of numerical magnitude representations in long-term memory* as a potential mediating factor [31, 73]. Evidence suggests that numbers are mentally represented on a spatially oriented linear continuum, the mental number line, with small/large numbers located on the left/right, respectively [78]. Accordingly, spatial language might provide a means to better grasp the spatial relations between numerical magnitudes on this mental continuum. The knowledge of spatial prepositions might enable children to better understand the spatial aspects of numerical quantity representations, which, in turn, could then positively affect their numerical skills. The precision of number-space mappings could thus be a plausible candidate for explaining the effects of spatial language on early numerical processing.

Nonetheless, findings from preverbal infants (e.g., [96]) and non-human animals [97] have advocated for a supposedly innate link between numbers and space, as they showed preferences for small/large nonsymbolic numerosities and the left/right side of space, respectively. Similar results were observed by Patro and Haman [98], showing directional spatial–numerical associations in precounting 4-year-old children using a nonsymbolic numerosity comparison task. These findings thus collectively suggest that the spatial mapping of numbers does not develop through formal schooling and is not specific to symbolic magnitudes. We thus believe that if spatial language indeed supported the spatial mapping of numerical magnitude representations on a long-term innate mental number line, it should beneficially relate to performances also on nonsymbolic numerical tasks, as indexed by numerosity comparisons in the present study. Unless one considers the possibility that mental number lines are notation-dependent and dissociable at the symbolic and nonsymbolic level [99]. In that case, one could envisage that spatial language only promotes the understanding of the spatial aspects of symbolic but not nonsymbolic quantity representations, thereby explaining the absence of a predictive effect on numerosity comparisons.

A second alternative explanation could be that spatial language facilitates the *spatial mapping of serial order in working memory* (WM). The maintenance of serial order in WM is supposedly achieved via position marking, with markers likely being of spatial nature considering systematic interactions between serial order in WM and space-related responses (e.g., [100, 101]).

Accordingly, memorized sequences are mentally stored on a spatially defined axis in both the verbal and visuospatial domains [102], where recall is supported by spatial attentional mechanisms efficiently scanning through the active item set [103]. In this view, one could hypothesize that the knowledge of spatial terms promotes the spatialization of ordered sequences in WM and/or the spatial scanning throughout, thereby offering an effective strategy for completing (numerical) tasks involving WM processing. The potential role of spatial language in the latter could then also account for its strong correlation with phonological awareness, repeatedly shown to associate with verbal WM (e.g., [104–107]).

Interestingly, counting was also shown to relate to verbal short-term memory (e.g., [108, 109]) and to depend on the retrieval of the correct order of number words [110, 111]. Actively recalling over-learnt count-list sequences from memory also seems essential when identifying adjacent ordered sequences as being in the correct ascending order and offers a plausible explanation for findings indicating faster responses when judging the order of adjacent (e.g., 1-2-3) compared to non-adjacent (e.g., 1-3-5) sequences [112]. Moreover, the ordinal understanding of symbolic numbers was shown to be related to individual differences in WM in adults [113]. Similar brain regions were also found to be activated when coding order information in WM tasks [114, 115] and when performing ordinal judgments in numerical tasks [116, 117]. Likewise, sequential memory seems required when classifying the magnitude of symbolic numbers with respect to that of a fixed referent, since the strength of number-space associations, measured in a symbolic magnitude classification task, was shown to depend on the availability of visuospatial WM resources [118]. Conversely, the ability to hold numerical information in correct order in WM seems less relevant when choosing the more numerous out of two simultaneously presented dot arrays or when mapping between visual and verbal number codes (i.e., transcoding). The influence of WM on number transcoding was indeed shown to be completely mediated by phonemic awareness [35]. This could then offer an explanation for why spatial language did not positively affect the two latter numerical skills, given the assumption that the spatialization of numerical information in WM acts as a mediating factor in the relation between spatial language and early numerical development.

A third and final explanation for the predictive effect of spatial language on symbolic number skills was provided by Gilligan-Lee et al. ([76], see also [73]). According to these authors, associations might be explained by *common requirements for grounding symbolic and conceptual representations*. To understand symbolic number, firstly number words and then Arabic digits have to be grounded with a conceptual understanding of numerical quantity [9, 10]. Likewise, grasping the meaning of verbal spatial terms depends on grounding these verbal labels with conceptual representations of spatial concepts [119]. As such, the ability to link symbolic numerical and spatial representations with their respective concepts would be a confounding factor in the relation between spatial language and symbolic numerical skills. Some children might simply be better at grounding symbolic with conceptual representations across domains. This could then explain the absence of a predictive relation between spatial language and numerosity comparisons in the present study.

It should, however, be noted that the neurocognitive integration between symbolic and nonsymbolic numbers has been seriously questioned (e.g., [11]), with recent evidence supporting the existence of distinct representations for numerosities and Arabic digits [120]. Moreover, Hurst et al. [121] showed that in preschool children Arabic digits are not directly mapped onto numerosities, but only indirectly tied to the quantities that they represent via the knowledge of number words. Accordingly, the ability to ground *verbal* symbols (i.e., number words and spatial terms) with conceptual representations across domains would be the confounding factor here.

### The relations of spatial abilities, phonological awareness and sociodemographic variables to early numerical skills before and after controlling for spatial language

Given the well-document relation between spatial and mathematical abilities across development ([37, 38], for reviews, see [36, 122]), we included *spatial abilities* as a potential predictor of early numerical skills in our analyses. To account for the multi-dimensionality of spatial abilities (for a review, see [80]), we assessed intrinsic-static and dynamic abilities in the figure ground task and CMTT, respectively, in addition to measuring extrinsic-static abilities in the spatial relations task, as it was previously done by Georges et al. [31] using a spatial perception task. Results show that spatial abilities significantly predicted both forward and backward counting, even when controlling for phonological awareness and the proficiency of the language of instruction (in step 2). This agrees with previous research highlighting the importance of spatial abilities even for early numerical skills of verbal nature (e.g., [30, 31, 40, 41, 44, 123, 124]). Interestingly, spatial abilities remained significant predictors of counting skills even when adding spatial language to the regression, suggesting that their effects are not fully mediated by this latter variable, as it was the case in Georges et al. [31]. One explanation for this discrepancy could be that Georges et al. [31] aggregated counting scores together with number naming into a single variable indexing verbal number skills. Since number naming was not predicted by spatial abilities (before or after controlling for spatial language) in the present study, this could explain the relatively weaker predictive effect of spatial perception in Georges et al. [31]. According to Zhang et al. [40], counting likely involves a multistep process requiring not only the verbal production of number words but also the ability to update (spatial) WM, especially when reciting backward sequences. Since the latter is also required when performing spatial tasks, this might explain the importance of spatial abilities for counting skills. This idea agrees with the aforementioned assumption that spatial language supports the development of numerical skills, including counting, via facilitating the spatial mapping of symbolic numbers in WM.

Interestingly, among the spatial tasks included in this study, spatial relations performances, indexing extrinsic-static abilities, were the only significant spatial predictor of counting skills. No additional variance was explained by intrinsic spatial abilities. One explanation for this could be that counting relies on the verbal coding of spatial relations between number words, similarly to how performances on extrinsic spatial tasks likely depend on the verbal mapping of spatial relations between objects. The assumption that verbal coding of spatial relations is used as a strategy not only during counting, but also extrinsic-static spatial tasks can be supported by the strong correlation between spatial relations performances and spatial language in the present study (see also [76], for stronger correlations between spatial language and extrinsic than intrinsic spatial tasks in older children).

Performances on the spatial relations task were also the only significant spatial predictor of numerosity comparisons, indexing approximate number system acuity, and of number writing. According to Zhang and Lin [125], early number knowledge likely relies to a lesser extent on sophisticated visuospatial abilities, such as mental rotation, as it does not require reasoning or transformation. It does, however, depend on linear numerical magnitude representations, whose scaling and precision might be affected by extrinsic-static spatial processes. This could thus explain the strong effect of the latter variable on numerosity comparisons. Conversely, the predictive effect of spatial relations performances on number writing might to some extent be explained simply by the fact that both tasks depended on visuomotor integration as they required children to either draw numbers (number writing task) or dot-connecting lines (spatial relations task).

As opposed to the aforementioned numerical tasks, symbolic magnitude classifications did not depend on extrinsic-static spatial abilities, but were significantly predicted by figure ground performances, falling into the intrinsic-static category. This might be surprising if one considers the processing of numerical relations as the cognitive mechanism underlying symbolic magnitude classifications, as this was just suggested to account for the relation between counting and extrinsic-static spatial abilities.

Finally, despite the significant correlations between the different spatial measures and ordinal judgments, spatial abilities were not predictive of this measure when controlling for the predictive effect of phonological awareness. According to Hutchison et al. [126], children's initial perception of ordinality is likely linked to the verbally rehearsed count-list. In other words, young children initially rely on verbal reciting strategies when judging numerical order. This could then explain young children's reliable performance when identifying adjacent ordered sequences as being in the correct ascending order as opposed to their struggle with non-adjacent ordered trials ([126], see also [112, 127]). A qualitative reorganization of how children process the ordinality of numerical sequences, caused by a shift from depending on a verbal list system to a spatially ordered representational system [128], then probably underlies their ability to extend notions of ordinality beyond the count-list at later developmental stages. In this vein, one might hypothesize that spatial abilities play a greater role in older children's ordinal processing skills, especially when judging more complex non-adjacent ordered sequences.

Apart from spatial abilities, we also considered the potentially predictive effect of *phonological awareness*, repeatedly shown to associate with mathematical knowledge, especially in younger children, even when controlling for domain-general factors (e.g., [29–34, 40, 129–131], for a meta-analysis, see [26]). Results show that it indeed significantly predicted early numerical skills, except for number writing, when controlling for spatial abilities in step 1 of the models. However, once controlled for the predictive effect of spatial language in step 3, it no longer related to forward counting, symbolic magnitude classifications or ordinal judgments, which were significantly predicted by spatial language instead. These findings thus point towards a greater importance of spatial language than phonological awareness for specifically those numerical skills. One could even hypothesize that spatial language acts as a full mediator in those instances in that better phonological awareness supports the acquisition of spatial language, which in turn promotes the development of forward counting as well as symbolic cardinal and ordinal judgments in young children. A similar relation was indeed observed in Georges et al. [31], indicating that spatial language fully mediated the predictive effect of phonological awareness (and spatial perception) on verbal number skills, an aggregate of counting and number naming abilities. Nonetheless, despite the present strong correlation between phonological awareness and spatial language, a causal relation cannot be confirmed. Especially a potentially confounding role of WM should be considered, given that verbal WM not only relates to phonological awareness (e.g., [104–107]), but might also affect or depend on spatial language (see suggested mechanisms accounting for the relation between spatial language and numerical skills in section above).

As opposed to forward counting, phonological awareness remained a significant predictor of backward counting even after controlling for the predictive effect of spatial language. This not only indicates that the ability to recognize and manipulate the spoken parts of sentences and words is especially relevant for operating the number chain backwardly, but also suggests that forward and backward counting likely depend on (partly) different cognitive processes [123, 132, 133]. According to Liu and Zhang [45], counting forward in a small familiar range might be relatively automatic, while backward counting might engage greater cognitive resources as it requires children to actively keep track of the ordinal sequence of number words when reciting them backwardly. In this vein, future studies might consider the effect of

verbal WM on backward counting and in explaining the strong relation between the latter variable and phonological awareness.

Finally, phonological awareness was the only significant predictor of number naming and also remained significantly associated with numerosity comparisons (alongside spatial abilities) after the inclusion of spatial language, which was not predictive in those instances. The former predictive effect might be expected, as brain regions involved in verbal processing and memory, notably the left angular gyrus, also underlie the manipulation of numbers in verbal form. Conversely, the latter relation seems surprising in light of the weak phonological representation hypothesis [28]. Accordingly, weak phonological skills should cause difficulties in numerical domains relying on verbal codes (e.g., counting, symbolic arithmetic) but spare other aspects of numeracy not directly depending on the manipulation of number words (e.g., subitizing, nonverbal calculation). As such, since numerosity comparisons likely reflect the acuity of the innate nonverbal approximate number system (e.g., [134]), one might not expect performances to depend on phonological skills. Nonetheless, some studies have argued that although nonsymbolic numerical representations are widely considering as being a precursor to symbolic numerical development (e.g., [9, 10]), including the acquisition of verbal number skills (see developmental model of [8]), the relation might also be reversed. According to the refinement hypothesis, experience with symbolic number processing enhances the precision of nonsymbolic number representations ([135], see also [136]). More specifically, it has been shown that only children already mastering verbal counting and able to use the role of cardinality, could discriminate nonsymbolic quantities independent of their perceptual properties, like overall size [137, 138]. This thus suggests that language, especially verbal counting knowledge, supports the development and sharpening of nonsymbolic number representation, thereby possibly explaining the predictive effect of phonological awareness in the present study.

As a final set of potential covariates, we considered the effects of the *sociodemographic variables* age, SES and language spoken with mother at home. In line with previous research, results show that children speaking Luxembourgish (the language of instruction) with their mother at home showed better phonological awareness (e.g., [139]) and spatial language skills [60] than their non-Luxembourgish speaking peers. Children speaking the native language at home also came from higher SES families, which agrees with the lower SES commonly associated with immigrant populations (e.g., [140]). Most importantly, Luxembourgish-speaking children also outperformed the second language learners on certain numerical measures [17–22], such as forward and backward counting as well as ordinal judgments. No differences between groups were observed on the number naming and writing as well as the symbolic magnitude classification and numerosity comparison tasks. The effects of instruction language proficiency were thus especially pronounced for numerical tasks depending on the count sequence knowledge. This is in line with the findings of Bonifacci et al. [20], indicating that preschool children with Italian as their second language underperformed native Italian speakers in numerical tasks with a verbal component, but not in supposedly nonverbal tasks such as quantity comparisons. Interestingly, the home language factor was still a significant predictor of forward and backward counting (but not ordinal judgments) when controlling for spatial abilities and phonological awareness in the regression models. It did, however, no longer affect any of these variables when additionally controlling for spatial language. One could thus hypothesize that spatial language fully mediated the relation between instruction language proficiency and counting abilities. This might thus be an interesting avenue for further investigations.

Apart from instruction language proficiency, higher SES also positively correlated with certain numerical measures [13, 14], such as forward counting, symbolic magnitude

classifications and ordinal judgments. It was, however, no longer related to forward counting or ordinal judgments when controlling for spatial abilities, phonological awareness and instruction language proficiency (in step 2), while it remained a significant predictor of symbolic magnitude classifications even after controlling for spatial language (in step 3). Interestingly, it also correlated with spatial language, which agrees with previous findings in 3-year-olds [71] as well as in preschool children, where it uniquely predicted the knowledge of spatial terms even when controlling for phonological awareness, spatial perception, age and gender (but see [73]).

## Implications

Given the importance of basic numerical abilities for later mathematical development [1, 2] beyond the effects of reading or attention skills [3], it is essential to identify children with relatively weaker numerical performances early in life to offer adequate training to thereby prevent future learning difficulties [141].

Since spatial language uniquely predicted early symbolic numerical skills in preschool, including verbal counting, magnitude classifications and ordinal judgments, one could envisage the design of spatial language tests to regularly screen young children for low performances to help identify those at risk for mathematical learning difficulties later in life. Such a spatial language measure, however, needs to be fairly short yet highly reliable to provide an adequate measure for making predictions about early numerical skills.

Moreover, training spatial language could be an interesting avenue for fostering numerical development in general and more specifically in children at risk for later learning difficulties prior to formal schooling. Based on the present findings, spatial language training might not only directly promote early numerical development, but also indirectly reduce (or even abolish) the negative impact of insufficient knowledge of the language of instruction on numerical skills, especially counting. It might also indirectly increase numerical performances via enhancing extrinsic-static spatial abilities (e.g., [58, 61]), shown here to correlate with spatial language and to predict performances on the vast majority of numerical tasks. Since spatial language correlated with SES and since children from lower SES backgrounds struggle when acquiring basic numerical skills [142, 143], training paradigms focused on promoting the use of spatial language might be especially fruitful in that instance.

With respect to possible training paradigms, one might for instance implement play activities where the orientation, placement, and spatial configuration of objects are constantly labelled, as this was previously shown to beneficially affect mental rotation skills in 4-year-old children [61]. Moreover, spatial assembly training with spatial language feedback might be a paradigm worth considering, as it was found to not only positively affect spatial but also numerical performances in three-year-olds, especially from lower SES backgrounds [55]. Alternatively, dialogic story book reading promoting the use of spatial terms might be a fruitful avenue, since a similar intervention focussing on mathematical language not only increased the latter but also general mathematical knowledge in 3- to 5-year-old children [69]. A final possibility is the use of semi-guided block play, since guided interactions with blocks were shown to not only naturally increase the use of spatial language in young children [144], but also positively affect numeracy skills in preschool [145].

## Limitations and future directions

Since this study was based on an associational design, we need to be careful not to interpret findings causally or to assume directionality of the observed relations. It could, for example, be possible that the development of symbolic numerical skills positively affects the

comprehension and use of mathematical language, including the knowledge of spatial terms (see [69]). However, we assume that it is more likely that the mastery of spatial language, starting to develop early in life, precedes and as such underlies early numerical skills in preschool. Some support for this assumption is provided by Lindner et al. [73], showing that spatial language in preschool predicted verbal number skills in first grade even when controlling for covariates such as vocabulary knowledge. Yet, future studies should further assess relations longitudinally or manipulate spatial language experimentally (e.g., through intervention; see also above) while focussing on a variety of numerical outcome measures to determine its causal importance in the development of early numerical skills.

Another important point worth mentioning is that the current study only focussed on locative prepositions, i.e., spatial terms within the category "locations and directions" [49]. The restriction to specifically these relation terms is in line with previous research (see e.g., [31, 71, 73, 74, 76]). It is also theoretically motivated based on potential cognitive mechanisms accounting for a link between spatial language and numerical cognition. However, future studies should assess how the knowledge of a broader range of spatial terms relate to each of the early numerical skills assessed in this study. Indeed, Pruden et al. [58] showed that one- to three-year-old children with a richer vocabulary to describe spatial features and properties of items (e.g., circle, triangle, corner, edge, line) showed higher performances on three non-verbal spatial problem-solving tasks at age 5. Given the importance of the latter tasks for a variety of numerical and mathematical skills (e.g., [36–38]), focussing on such a broader range of spatial language might thus further highlight its importance for mathematical development. Interestingly, Hawes et al. [72] reported positive correlations between the comprehension of location terms, shape and figure names and numeracy outcomes, such as nonsymbolic and symbolic number comparisons. Nonetheless, it remains to be determined whether the knowledge of spatial terms extending beyond the category "locations and directions" uniquely predicts numerical cognition or only indirectly positively affects numerical development via an effect on e.g., spatial abilities.

Future studies should also control for the effects of quantitative language when assessing relations between spatial language and numerical knowledge. Most existing studies have either focussed on mathematical language, intermingling quantitative and spatial terms, or focussed exclusively on spatial language (but see [75]). Given strong correlations between spatial and quantitative languages [75], one should control for the latter when aiming at determining the unique effects of the former on early mathematical development.

Moreover, future research should replicate the current findings with different and/or more reliable (i.e., Cronbach's $\alpha > .70$) spatial measures to determine whether the main conclusions hold when using other indices of intrinsic-static (e.g., the Children's Embedded Figures Task, CEFT) and intrinsic-dynamic (e.g., the mental folding task, [146]) as well as extrinsic-static (e.g., the copying subtest of the FEW-II) spatial abilities. One should also additionally focus on the potentially predictive effect of extrinsic-dynamic spatial abilities [76], as indexed by perspective taking performances [147], once this skill is fully mastered in older children [147].

In this vein, older age groups throughout elementary school should also be considered to determine relations between spatial language and more complex arithmetic and geometry problems both concurrently and longitudinally. To the best of our knowledge, there is only one study so far assessing the predictive effect of spatial language on general mathematical abilities in children aged 6 to 10 years [76].

Future studies should also further unravel the cognitive mechanisms underlying spatial language as well as its relation to early numerical skills. Here, spatial language correlated with phonological awareness as well as intrinsic-dynamic and extrinsic-static spatial abilities, while no relation was observed for performances on the figure ground task. Future research should

focus on the relative contributions of these variables when simultaneously controlling for home language, SES background, and possibly also visuospatial and verbal WM. The latter cognitive variables should also be considered when further assessing the mental processes explaining relations between spatial language and certain (but not all) aspects of early numerical development (e.g., counting). To get a better understanding of the mechanisms underlying the predictive effects of spatial language, one might also, for instance, specifically contrast relations between spatial language and ordinal judgments using adjacent as well as non-adjacent ordered sequences in younger and older children, since only the former but not the latter age group supposedly still relies on the verbal count list when judging numerical order (e.g., [127]).

Finally, it is important to remind that the present findings might only apply to typical young children without learning disabilities (e.g., dyscalculia) and/or sensory impairments (e.g., deafness). Future studies should extend the present findings by determining whether the current pattern of results also holds in atypical populations, such as for instance deaf children, who differ in their spatial abilities and reliance on phonological awareness skills.

## Conclusion

In the present study, we focussed on preschool children to determine the effects of spatial language, as indexed by the production and comprehension of locative prepositions, on early numerical skills, such as counting, number naming and writing, symbolic magnitude classifications and ordinal judgments as well as numerosity comparisons. We controlled for the potentially confounding effects of different spatial abilities (accounting for their multidimensionality), phonological awareness, age, SES as well as home language and as such indirectly instruction language proficiency.

Spatial language significantly predicted both forward and backward counting alongside extrinsic-static spatial abilities as well as phonological awareness in the latter case. Conversely, number naming, previously aggregated together with counting skills into a single score indexing verbal number skills, did not depend on spatial language but was solely predicted by phonological awareness. Likewise, number writing only depended on extrinsic-static spatial abilities. Spatial language did also not affect numerosity comparisons. It did, however, significantly predict symbolic magnitude classifications and was the only predictor of ordinal judgments. These findings thus highlight the unique importance of spatial language for early numerical development beyond verbal number skills and suggest that the knowledge of spatial terms might be especially relevant for numerical tasks depending on the verbal coding of spatial relations between symbolic numbers.

Given the unique importance of spatial language for early symbolic numerical development, future studies aimed at better understanding the cognitive mechanisms underlying spatial language as well as its relation to different numerical skills should help in the design of adequate training paradigms for fostering early mathematical knowledge.

## Acknowledgments

We would like to thank Ms Tânia Ramos for her invaluable help in designing the ordinal judgment task and in supervising data collection. We would also like to thank Dr Mathieu Guillaume for providing the script of the numerosity comparison task.

## Author Contributions

**Conceptualization:** Carrie Georges, Véronique Cornu, Christine Schiltz.

**Data curation:** Carrie Georges.

**Formal analysis:** Carrie Georges.

**Funding acquisition:** Christine Schiltz.

**Methodology:** Carrie Georges, Christine Schiltz.

**Supervision:** Christine Schiltz.

**Writing – original draft:** Carrie Georges.

**Writing – review & editing:** Carrie Georges, Véronique Cornu, Christine Schiltz.

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
