## [Decision Letter · Decision Letter 0]

6 Jul 2023

PONE-D-23-11288The importance of spatial language for early numerical development in preschool: Going beyond verbal number skillsPLOS ONE

Dear Dr. Georges,

Thank you for submitting your manuscript to PLOS ONE. After careful consideration, we feel that it has merit but does not fully meet PLOS ONE’s publication criteria as it currently stands. Therefore, we invite you to submit a revised version of the manuscript that addresses the points raised during the review process. Please respond to the comments from the reviewer and I would appreciate your thoughts on the question I asked about deaf individuals, math and phonological awareness. Please submit your revised manuscript by Aug 20 2023 11:59PM. If you will need more time than this to complete your revisions, please reply to this message or contact the journal office at plosone@plos.org. Please include the following items when submitting your revised manuscript:A rebuttal letter that responds to each point raised by the academic editor and reviewer(s). You should upload this letter as a separate file labeled 'Response to Reviewers'.A marked-up copy of your manuscript that highlights changes made to the original version. You should upload this as a separate file labeled 'Revised Manuscript with Track Changes'.An unmarked version of your revised paper without tracked changes. You should upload this as a separate file labeled 'Manuscript'.

We look forward to receiving your revised manuscript.

Kind regards,

Mary Diane Clark, PhD

Academic Editor

PLOS ONE

Additional Editor Comments:

Thank you for submitting your manuscript. Summer is a difficult time to get multiple reviewers so I am going forward with my own feedback and reviewer 1's feedback. Please address the reviewers concerns about reliability as well as the other comments.

I do have a question from my own read about phonological awareness --- for many deaf children phonological awareness is not necessary in reading. Math abilities is related to their visual spatial ability. Might you discuss this as a limitation in your revision?

Reviewers' comments:

Reviewer's Responses to Questions

**Comments to the Author**

1. Is the manuscript technically sound, and do the data support the conclusions?

Reviewer #1: No

2. Has the statistical analysis been performed appropriately and rigorously? 

Reviewer #1: No

3. Have the authors made all data underlying the findings in their manuscript fully available?

Reviewer #1: Yes

4. Is the manuscript presented in an intelligible fashion and written in standard English?

Reviewer #1: Yes

5. Review Comments to the Author

Reviewer #1: The authors outline a study that examined the numerical skills in preschool-aged children uniquely predicted by spatial language may uniquely predict early numeracy skills in preschool-aged children. One hundred fifty-five children, ranging in age from 5 ½ to 7 ½ years, were assessed on their comprehension and ability to name location terms, their numerical skills, spatial skills, and phonological awareness. Parents reported on their socioeconomic status and the predominant language spoken to their child by each parent. About half of the children spoke Luxembourgish with their mother at home, while the remaining children in the sample spoke a distinct language with their mother, several did speak Luxembourgish with their father, and those that did not speak Luxembourgish with either parent but did hear this language at school. Controlling for family socioeconomic background, phonological awareness, and spatial abilities, children’s spatial language predicted several numerical abilities, including forward and backward counting, magnitude comparisons and symbolic ordinal judgments. The authors propose that encouraging preschool children’s spatial language may be a viable pathway for fostering the development of numerical skills.

The authors report an intriguing and exciting relation between children’s spatial language and a broad range of numerical skills. The present results extend the relation between spatial language and numerical skills to a broader array of numerical skill than previously documented. The present study also includes a linguistically diverse sample of children, including those whose predominant language is distinct from the language of instruction, Luxembourgish. In sum, there are several noteworthy aspects of the present study.

The potential contribution of the present results was dampened by several concerns with the internal reliability of the spatial tasks and questions about focusing the present study on children’s location terms. The inclusion of a Cronbach’s alpha for each measure was greatly appreciated. This measure of internal consistency of each spatial task was below the level argued to be within the acceptable range, above .70. None of the three spatial tasks approached this level of internal consistency, raising questions of whether children demonstrated consistent responses across items of each measure. The adoption of abbreviated versions of the CMTT may also account for the lower Cronbach’s alpha. The concern with the Cronbach’s alpha for each spatial task is that these spatial tasks may not have tapped children’s spatial skill effectively. The relation between spatial language and numerical abilities reported in the present study might not hold if other measures of spatial skills had been adopted in the study.

In addition to concerns about the reliability of the spatial tasks (and thus, their construct validity), the introduction did not provide a clear rationale for focusing on location terms in the present study. Rather, the authors provide a general overview of the link between spatial language and mathematical in broad terms. Specifying the types of spatial language examined in each previous study will better situate the present study in the context of previous work. This information may also better motivate the rationale for focusing on children’s location terms. Although some studies have focused on these spatial words, they did so in relation to a spatial task closely related to the semantics of those spatial terms (see Balcomb et al., 2012; Piccardi et al., 2015). Because the goal of the present study was to examine a broad range of numerical skills, the focus on location spatial terms was not clear. Might the present results generalize to a broader range of spatial language? Might the results be even more robust with spatial words that have been linked to spatial thinking (see Pruden et al., 2011)?

Finally, the authors control for age in their analyses but in their table, report age for only 154 of their 155 participants. Was this value missing? How was the data for the participant with missing age managed in analyses that controlled for age?

6. PLOS authors have the option to publish the peer review history of their article (what does this mean?). If published, this will include your full peer review and any attached files.

Reviewer #1: No

---

## [Author Response · Author response to Decision Letter 0]

3 Aug 2023

Dear Editor, Prof. Dr Mary Diane Clark, 

Dear Reviewer,

We would like to thank the Editor and the Reviewer for their thoughtful comments and suggestions. We addressed each concern in a revision of our manuscript. All changes to its original version are highlighted by track-changes and point-by-point answers are provided below (please see also "Response to Reviewers" file). 

Editor Comments:

1. I do have a question from my own read about phonological awareness – for many deaf children phonological awareness is not necessary in reading. Math abilities are related to their visual spatial ability. Might you discuss this as a limitation in your revision?

Thank you for this very interesting comment. We have now included a paragraph in the limitations and future directions section stating that the current findings might only apply to healthy young children. Future studies should assess the importance of each of the factors included in the present study, notably spatial language and phonological awareness, for numerical development in populations with sensory impairments like deafness. The paragraph reads as follows:

“Finally, it is important to remind that the present findings might only apply to healthy young children without learning disabilities (e.g., dyscalculia) and/or sensory impairments (e.g., deafness). Whether or not the current pattern of results stressing the crucial role of linguistic factors like spatial language and phonological awareness in early numerical development in preschool can be generalized to young children not using a conventional spoken language due to very little or no functional hearing should be assessed in future studies. 

Notably, previous studies have consistently shown that deaf and hard-of-hearing children, not exposed to fluent sign language from birth, generally show severe and persistent difficulties in numerical, arithmetical and mathematical skills compared to their hearing peers from preschool onward [148-154]. Especially, tasks requiring symbolic as opposed to nonsymbolic quantity processing seem more challenging (e.g., [155]). Their delay also appears most pronounced on verbal numerical tasks, including counting [156], fractions [157], problem solving [158], and arithmetic skills, notably multiplications (e.g., [159]), compared to visuospatial tasks, such as geometry (e.g., [158]). In line with the current findings, the difficulties of deaf pupils were assumed to be due amongst others to their lower language abilities early in development and/or their lack of exposure to spontaneous numerical language in their home environment (for a review, see [160]). Importantly, despite their language impairment, deaf individuals were previously reported to show distinct advantages in various visual domains, including visual attentional shifting and scanning [161], peripheral detection of motion (e.g., [162]), as well as the manipulation of mental images (e.g., [163]). They were also reported to benefit from instructions emphasizing spatial representations and strategies when solving numerical tasks. Namely, Zarfaty et al. [164] showed that 3- to 4-year-old deaf children outperformed their hearing peers on a task requiring the recollection of the number of items in a set when presented simultaneously in a spatial array but not when displayed sequentially in a temporal sequence. Similarly, Chen and Wang [165] showed that general cognitive abilities such as spatial abilities and processing speed were better predictors of mathematical performances in deaf or hard-of-hearing students than nonsymbolic and symbolic numerical magnitude processing, thereby likely implying the particular importance of spatial strategies for mathematical achievement in those individuals. Promoting the use of such strategies in deaf children might thus be more beneficial for early numerical development and yield greater gains than stressing the use of spatial language and/or phonological awareness in that population. 

Nonetheless, the relative contributions of spatial language, phonological awareness and spatial abilities to early numerical development in deaf children needs further investigation. More concretely, it should be assessed whether or not deaf or hard-of-hearing individuals are in fact true visual learners (but see, [166]) in that most of the early numerical abilities included in the present study are actually predicted by spatial abilities rather than phonological awareness and/or spatial language. One should also determine if potential performance differences of deaf pupils with respect to their hearing peers could then be explained by their preferential use of visuospatial strategies on numerical tasks primarily predicted by verbal factors, such as spatial language, in healthy children. Notably, Buyle and colleagues [159] recently speculated that the lower performances of deaf individuals particularly on multiplication but not subtraction problems might be due to their reliance on a visuospatial as opposed to verbal route when solving multiplications. This visuospatial detour might impose a greater cognitive load and thereby explain slower performances in deaf individuals”. 

Reviewers' comments:

1. The potential contribution of the present results was dampened by several concerns with the internal reliability of the spatial tasks and questions about focusing the present study on children’s location terms. The inclusion of Cronbach’s alpha for each measure was greatly appreciated. This measure of internal consistency of each spatial task was below the level argued to be within the acceptable range, above .70. None of the three spatial tasks approached this level of internal consistency, raising questions of whether children demonstrated consistent responses across items of each measure. The adoption of abbreviated versions of the CMTT may also account for the lower Cronbach’s alpha. The concern with the Cronbach’s alpha for each spatial task is that these spatial tasks may not have tapped children’s spatial skill effectively. The relation between spatial language and numerical abilities reported in the present study might not hold if other measures of spatial skills had been adopted in the study.

Thank you for this comment. We absolutely agree with the Reviewer that the internal reliabilities, as indexed by Cronbach’s alpha, of the three spatial tasks were all below .70, which can be considered as a cut-off value for acceptability. Nonetheless, according to some authors (e.g., DeVellis, 2003; Konting et al. 2009; Nunnally, 1978), values between .60 and .69 also fall within the acceptable range. Accordingly, values within this range were previously acknowledged to be “acceptable”, “adequate”, “good”, “medium”, “moderate”, “reasonable”, and “satisfactory” by a variety of authors with respective measurements then also included in subsequent analyses (e.g., Gilligan-Lee et al., 2021; Koponen et al., 2016; Mussolin et al., 2014; Quaiser-Pohl et al., 2014; see also Taber, 2018, for a summary). As such, we believe that there is no clear consensus in the literature on the threshold of acceptability for Cronbach’s alpha values and that cut-off values are therefore still a matter of debate.

When more specifically screening the literature for studies that either used any of the spatial tasks also administered in the current study (i.e., figure ground and spatial relations tasks of the FEW-II, or CMTT) and/or assessed the effects of spatial language on numerical cognition while controlling for spatial abilities, comparisons are difficult since only a minority of these studies have reported internal consistency measures. For instance, Lindner et al. (2022) administered the visual-perception subtest of the Beery-Buktenica Developmental Test of Visual-Motor Integration as a covariate in the relation between spatial language and verbal number skills and indicated a split-half reliability based on the odd-even method of r = .329. In a related study, these authors reported a Cronbach’s alpha of .67 for a mental rotation task adapted from Gilligan et al. (2019). With respect to the CMTT, Cronbach’s alpha values in previous studies ranged from .43 to .70. Casasola et al. (2020) administered two 10-item versions of this task and indicated Cronbach’s alpha values of .57 and .54. Hawes et al. (2015) used the 16-item version also included in the current study and reported a split-half reliability based on the odd-even method of r = .69 and a Cronbach’s alpha of .70. Cornu et al. (2017) indicated a Cronbach’s alpha value of .69 for their shortened 12-item version. The 10-item version of Turan et al. (2023) only yielded a Cronbach’s alpha of .43, while that of Xu and LeFevre (2016) was .61. As opposed to the CMTT, the figure ground and spatial relations tasks were taken from a standardized test battery, the FEW-II (“Frostigs Entwicklungstest der visuellen Wahrnehmung 2”, Second Edition), which is the German equivalent of the Developmental Test of Visual Perception, Second Edition (DTVP-2). The mean values for Cronbach's alpha were reported to be acceptable for all subtests, ranging between .67 and .92. With values of .85 and .80 for the figure ground and spatial relations task, respectively, internal reliabilities were indeed higher in the study of Meinhardt et al. (2021) compared to ours. 

To determine whether the current study’s main conclusions hold when trying to increase internal consistencies of the different spatial tasks, we ran some additional analyses. 

First of all, based on item statistics, notably “Cronbach’s alpha if item deleted” and “corrected item-total correlation”, we removed 3 and 2 items from the figure ground task and CMTT, respectively. No items had to be removed from the spatial relations task based on these criteria. Cronbach’s alpha only slightly improved from .61 to .62 and from .65 to .67 for the figure ground task and CMTT, respectively. Removing these items did not change any of the regression outcomes. 

Secondly, we computed a composite score by adding all the items of each of the three spatial tasks. In that case, Cronbach’s alpha was .76. Including this more reliable spatial measure in our regression analyses did, however, not change any of the outcomes except that spatial abilities were now also a significant predictor of symbolic ordinal judgments alongside spatial language in the final step of the model. For the sake of completeness, we also computed split-half reliabilities based on the odd-even method for each spatial measure and included the Spearman-Brown coefficients in the method section: figure ground: r = .66, CMTT: r = .69 (see same value in Hawes et al., 2015), and spatial relations: r = .76. 

Concerning task length, we totally agree with the Reviewer that one way to probably improve the internal reliability of the CMTT would be to administer the longer 32-item version, initially proposed by Levine et al. (1999). Albeit, to the best of our knowledge, neither the original study nor any studies subsequently using this longer version reported any reliability statistics. The original task consisted of 16 items requiring mental rotation and 16 items involving mental translation (i.e., mentally moving the two pieces without rotating them – a simpler task). Considering that we used this task to index intrinsic-dynamic spatial abilities (i.e., mental rotation), we opted to include only those items requiring mental rotation, as in Hawes et al. (2017). This information has now been added to the method section.

Considering the Reviewer’s concern that the current findings might not hold when using different and/or more reliable spatial measures, it should be noted that similar results stressing the importance of spatial language for early mathematical development were also previously reported when using different spatial tasks as covariates. 

•For instance, Gilligan-Lee et al. (2021) observed positive relations between spatial language and standardized mathematics skills as well as estimations on 0-100 number lines, after controlling, amongst others, for spatial abilities, as indexed by the Children’s Embedded Figures Task (CEFT), a mental rotation task (as in Broadbent, Farran, & Tolmie, 2014), a spatial scaling task (as in Gilligan, Hodgkiss, Thomas, & Farran, 2018) and a perspective taking task (as in Frick, Möhring, & Newcombe, 2014). 

•Likewise, the use of egocentric reference frames in spatial language in 4- to 5-year-old children was significantly related to their number word comparison performances, even when accounting for mental rotation skills (indexed as in Gilligan et al., 2019). 

•Georges et al. (2021) as well as Lindner et al. (2022) reported significant effects of spatial language on verbal number skills, as indexed by aggregating scores on counting and number naming tasks, when controlling for spatial perception skills, as measured by an adaptation of the position in space subtest of the Developmental Test of Visual Perception (DVTP-2; see also Cornu et al., 2018) and the visual-perception subtest of the Beery-Buktenica Developmental Test of Visual-Motor Integration, respectively. 

•Finally, Bower et al. (2020) showed that the comprehension of spatial relation terms (e.g., under, above) in 3-year-olds mediated the relation between spatial abilities, as measured with the 2D and 3D TOSA (Verdine et al., 2014), and early numerical skills, especially in girls. Moreover, it should be noted that spatial abilities, mostly spatial relations, were a significant predictor of most numerical outcome measures. 

As such, we do not believe that relations between spatial language and the currently assessed numerical skills were merely due to a lack of predictive effect of spatial abilities, as a consequence of their relatively lower internal reliabilities. Nonetheless, we have now added a paragraph in the limitations and future directions section, stating that future studies should replicate the current findings when using different and/or more reliable spatial measures. The paragraph reads as follows:

“Moreover, future research should replicate the current findings with different and/or more reliable (i.e., Cronbach’s α > .70) spatial measures to determine whether the main conclusions hold when using other indices of intrinsic-static (e.g., the Children’s Embedded Figures Task, CEFT) and intrinsic-dynamic (e.g., the mental folding task, [146]) as well as extrinsic-static (e.g., the copying subtest of the FEW-II) spatial abilities. One should also additionally focus on the potentially predictive effect of extrinsic-dynamic spatial abilities [76], as indexed by perspective taking performances [147], once this skill is fully mastered in older children [147]”.

2. In addition to concerns about the reliability of the spatial tasks (and thus, their construct validity), the introduction did not provide a clear rationale for focusing on location terms in the present study. Rather, the authors provide a general overview of the link between spatial language and mathematics in broad terms. Specifying the types of spatial language examined in each previous study will better situate the present study in the context of previous work. This information may also better motivate the rationale for focusing on children’s location terms. Although some studies have focused on these spatial words, they did so in relation to a spatial task closely related to the semantics of those spatial terms (see Balcomb et al., 2012; Piccardi et al., 2015). Because the goal of the present study was to examine a broad range of numerical skills, the focus on location spatial terms was not clear. Might the present results generalize to a broader range of spatial language? Might the results be even more robust with spatial words that have been linked to spatial thinking (see Pruden et al., 2011)?

We totally agree with the Reviewer that we did not provide a clear explanation for why we only considered spatial relation terms as an index of spatial language proficiency in the present study. We have now added a rationale for specifically focussing on locative prepositions, i.e., spatial terms within the category “locations and directions”, in the section “the present study”. The paragraph reads as follows:

“Since most previous studies investigating the role of spatial language in numerical cognition have considered locative prepositions (see e.g., [31, 71, 73, 74, 76]), i.e., spatial terms within the category “locations and directions” [49], the present study also specifically focussed on the latter. This category also seems most relevant when considering that numerical magnitudes are supposedly mentally represented on a spatially oriented linear continuum, the mental number line, with small/large numbers located on the left/right, respectively [78]. Numbers can thus be spatially localized in relation to each other depending on their cardinality and/or ordinality (e.g., 2 is before, next to, on the left of 3). Since the spatial mapping of numerical magnitude representations seems like a plausible candidate through which spatial language could facilitate numerical cognition, assessing the knowledge of spatial relation terms seems most pertinent in this case (see also [73] for a similar rationale)”.

We have now also provided further information about the specific spatial terms assessed in each of the studies mentioned in the manuscript. Please see paragraph:

“Importantly, spatial language was shown to not only positively associate with spatial abilities (e.g., [58, 61]) but to also relate to numerical skills in young children (for a recent review, see [62]). The knowledge of spatial terms thus seems to facilitate spatial cognition more broadly (for a review, see [63]) by also affecting the spatial aspects of numerical processing. For instance, mathematical language, assessed using Purpura and Logan’s [64] measure of mathematical content language including six quantitative items (i.e., take away, a little bit, more, less, most, and fewest) and ten spatial items (i.e., nearest, under, first, far, below, front, middle, end, last, and before), was found to predict a variety of numeracy outcomes, over and above general language skills [56, 65]. The same measure was also recently shown to account for the relation between home numeracy environment (i.e., home activities such as counting objects) and children’s mathematical abilities [66]. Mathematical language, indexed with 24 quantitative (e.g., half, equal, more) and spatial (e.g., behind, between, opposite) items from the “sentence structures” subtest of the Language Test for All Children [67], was also shown to mediate the relation between general language and numerical development in preschool [68]. Moreover, training mathematical language (focussing on both quantitative and spatial terms) via story book reading in 3- to 5-year-olds was shown to improve general mathematical knowledge compared to a business-as-usual control group ([69] but see [70]). Furthermore, when focussing more specifically on spatial language, Bower et al. [71] reported that the comprehension of spatial relation terms (i.e., under, above, between, up, in, on, down, behind, below, middle, in front of, next to, on top of, and upside down) in 3-year-olds mediated the relation between spatial abilities and early numerical skills, especially in girls. Hawes et al. [72] also indicated significant correlations between spatial language comprehension and numeracy outcomes in 6-year-olds. In that study, children were required to identify their left vs. right hand, the location of an object in relation to a box, and various shapes and figures (e.g., cube). Moreover, Georges et al. [31] found that preschool children’s spatial language skills, assessed by the production and comprehension of locative prepositions (i.e., on, left, before, in, right, behind, above, and under), predicted their verbal number skills, reflecting performances on both counting and number naming tasks, even when accounting for the influences of phonological awareness, spatial perception as well as age, gender, and socioeconomic status. The importance of spatial language, as assessed in Georges et al. [31], for verbal number skills in young children was also confirmed by Lindner et al. [73], after controlling for potentially confounding variables such as spatial perception, vocabulary knowledge, age and gender. In their study, spatial language even predicted the acquisition of verbal numerical skills 6 months later, thereby further highlighting the importance of the mastery of spatial relation terms in the development of verbal number knowledge. This also agrees with the observation that in 4- to 5-year-old children the knowledge of the spatial terms “in front”, “left”, “right”, and “behind”, when used in an egocentric reference frame, significantly related to their number word comparison performances, even when accounting for socioeconomic background, mental rotation skills, as well as the understandings of absolute magnitude and numerical sequences [74]. Turan and colleagues [75] recently extended these findings by showing that spatial language, as assessed by a measure focussing on 12 spatial items (i.e., far away, at the end, front, closest to, on, first, last, under, behind, in front of, under, middle) adapted from Purpura and Logan [64], was significantly associated with preschool children’s geometry performances and also partially mediated the effects of general language on the latter. Finally, Gilligan-Lee et al. [76] observed positive relations between spatial language and standardised mathematics skills as well as estimations on 0-100 number lines but not dot comparison performances in slightly older children aged 6 to 10 years, after controlling for spatial abilities, vocabulary knowledge, age, and grade. To account for the slightly older age range of the children, they adapted the spatial language measure of Farran and Atkinson [77] to include the following 12 spatial relation terms: above, below, right, left, between, around, through, higher, lower, closer, further, parallel”. 

We have now also discussed the current study’s restriction to only locative prepositions (i.e., spatial relation terms) in the limitations and future directions section and suggest that the importance of a broader range of spatial terms for early mathematical development should be assessed in further studies. The added paragraph reads as follows:

“Another important point worth mentioning is that the current study only focussed on locative prepositions, i.e., spatial terms within the category “locations and directions” [49]. The restriction to specifically these relation terms is in line with previous research (see e.g., [31, 71, 73, 74, 76]). It is also theoretically motivated based on potential cognitive mechanisms accounting for a link between spatial language and numerical cognition. However, future studies should assess how the knowledge of a broader range of spatial terms relate to each of the early numerical skills assessed in this study. Indeed, Pruden et al. [58] showed that one- to three-year-old children with a richer vocabulary to describe spatial features and properties of items (e.g., circle, triangle, corner, edge, line) showed higher performances on three non-verbal spatial problem-solving tasks at age 5. Given the importance of the latter tasks for a variety of numerical and mathematical skills (e.g., [36-38]), focussing on such a broader range of spatial language might thus further highlight its importance for mathematical development. Interestingly, Hawes et al. [72] reported positive correlations between the comprehension of location terms, shape and figure names and numeracy outcomes, such as nonsymbolic and symbolic number comparisons. Nonetheless, it remains to be determined whether the knowledge of spatial terms extending beyond the category “locations and directions” uniquely predicts numerical cognition or only indirectly positively affects numerical development via an effect on e.g., spatial abilities”. 

3. Finally, the authors control for age in their analyses but in their table, report age for only 154 of their 155 participants. Was this value missing? How was the data for the participant with missing age managed in analyses that controlled for age?

Thank you for this comment. Information about age was indeed missing for one child. We have now added the following information to the method section under the heading “Covariates”:

“Age was calculated by averaging the children’s ages across both testing sessions. Information about age, HISEI and language background was missing for one, seven and six children, respectively. These children were excluded from all analyses including one or more of those covariates.”

Importantly, children with missing descriptives (i.e., information about age, SES, and/or language background) did not feature particularly low or high spatial languages or numerical skills (see Table in "Response to Reviewers" file).

---

## [Decision Letter · Decision Letter 1]

14 Sep 2023

PONE-D-23-11288R1The importance of spatial language for early numerical development in preschool: Going beyond verbal number skillsPLOS ONE

Dear Dr. Georges,

Thank you for submitting your manuscript to PLOS ONE. After careful consideration, we feel that it has merit but does not fully meet PLOS ONE’s publication criteria as it currently stands. Therefore, we invite you to submit a revised version of the manuscript that addresses the points raised during the review process.

Please see the section that the reviewer and I would like to see deleted.  Other changes were fine.

We look forward to receiving your revised manuscript.

Kind regards,

Mary Diane Clark, PhD

Academic Editor

PLOS ONE

Journal Requirements:

Additional Editor Comments:

I am attaching what the reviewer and I think you should take out of this version. Sorry my suggestion got you into a rabbit hole. There are other issues and I would love to see this done with young deaf children by a researcher who is aware of the current publications but some of. your statements disagree with these new publications.

so if you take out what is in the attachment we should be ready to go

Reviewers' comments:

Reviewer's Responses to Questions

**Comments to the Author**

1. If the authors have adequately addressed your comments raised in a previous round of review and you feel that this manuscript is now acceptable for publication, you may indicate that here to bypass the “Comments to the Author” section, enter your conflict of interest statement in the “Confidential to Editor” section, and submit your "Accept" recommendation.

Reviewer #2: All comments have been addressed

2. Is the manuscript technically sound, and do the data support the conclusions?

Reviewer #2: Yes

3. Has the statistical analysis been performed appropriately and rigorously? 

Reviewer #2: Yes

4. Have the authors made all data underlying the findings in their manuscript fully available?

Reviewer #2: Yes

5. Is the manuscript presented in an intelligible fashion and written in standard English?

Reviewer #2: Yes

6. Review Comments to the Author

Reviewer #2: Given the length of this manuscript (the download was 155 pages) and the short time provided to review it (10 days), I am going to presume that the expectation was to see if the queries of prior reviewers were met. If this was the purpose of an additional review, then indeed, questions brought up by prior reviewers seem to be met by the author.

In this reviewer’s opinion however, the manuscript seems excessively long and arduous to read. Several concepts that appear to be tangentially related (e.g., phonological awareness, spatial awareness, number mapping, language exposure) that would likely come together in the course of normal child development are studied for their relation to one another making the manuscript appear to lack true direction and purpose. The reference to deaf children and the statement that deaf learners may not truly be visual learners (page 49) seems out of place and not grounded in any finding that would emerge from the research in this study. This study did not include deaf children so statements of this nature cannot be made.

The manuscript would benefit from being drastically cut and streamlined- if the topic of focus is the relationship between children’s use of spatial language and how this related to their early numerical skills, then focus on this and only this.

7. PLOS authors have the option to publish the peer review history of their article (what does this mean?). If published, this will include your full peer review and any attached files.

Reviewer #2: No

---

## [Author Response · Author response to Decision Letter 1]

15 Sep 2023

Dear Editor, Prof. Dr Mary Diane Clark, 

Dear Reviewer,

Thank you for the positive evaluation of the revision of our manuscript. Our responses are in purple.

Additional Editor Comments:

Please see the section that the reviewer and I would like to see deleted. Other changes were fine.

I am attaching what the reviewer and I think you should take out of this version. Sorry my suggestion got you into a rabbit hole. There are other issues and I would love to see this done with young deaf children by a researcher who is aware of the current publications but some of your statements disagree with these new publications.

So, if you take out what is in the attachment, we should be ready to go.

Thank you for this feedback. We have now deleted the paragraph about deaf individuals, as indicated in the attachment. As suggested by the Editor, we have now only briefly introduced the topic of deafness in children by concluding the “Limitations and future directions” section with the following:

“Finally, it is important to remind that the present findings might only apply to typical young children without learning disabilities (e.g., dyscalculia) and/or sensory impairments (e.g., deafness). Future studies should extend the present findings by determining whether the current pattern of results also holds in atypical populations, such as for instance deaf children, who differ in their spatial abilities and reliance on phonological awareness skills”. 

If the Editor would prefer us to remove this short paragraph as well, please let us know.

Reviewer Comments:

Given the length of this manuscript (the download was 155 pages) and the short time provided to review it (10 days), I am going to presume that the expectation was to see if the queries of prior reviewers were met. If this was the purpose of an additional review, then indeed, questions brought up by prior reviewers seem to be met by the author.

Thank you for this positive feedback.

In this reviewer’s opinion however, the manuscript seems excessively long and arduous to read. Several concepts that appear to be tangentially related (e.g., phonological awareness, spatial awareness, number mapping, language exposure) that would likely come together in the course of normal child development are studied for their relation to one another making the manuscript appear to lack true direction and purpose. The reference to deaf children and the statement that deaf learners may not truly be visual learners (page 49) seems out of place and not grounded in any finding that would emerge from the research in this study. This study did not include deaf children so statements of this nature cannot be made.

The manuscript would benefit from being drastically cut and streamlined - if the topic of focus is the relationship between children’s use of spatial language and how this related to their early numerical skills, then focus on this and only this.

As also suggested by the Editor, we have now removed the paragraph about deaf individuals in the “Limitations and future directions” section to make this manuscript more focussed on the importance of spatial language for early numerical skills in typical child development.

---

## [Editor Report · Decision Letter 2]

18 Sep 2023

The importance of spatial language for early numerical development in preschool: Going beyond verbal number skills

PONE-D-23-11288R2

Dear Dr. Georges,

We’re pleased to inform you that your manuscript has been judged scientifically suitable for publication and will be formally accepted for publication once it meets all outstanding technical requirements.

Kind regards,

Mary Diane Clark, PhD

Academic Editor

PLOS ONE

Additional Editor Comments (optional):

Thank you for these efforts. Spatial languages and math are related in many interesting ways.
---

## [Editor Report · Acceptance letter]

22 Sep 2023

PONE-D-23-11288R2 

The importance of spatial language for early numerical development in preschool: Going beyond verbal number skills 

Dear Dr. Georges:

I'm pleased to inform you that your manuscript has been deemed suitable for publication in PLOS ONE. Congratulations! Your manuscript is now with our production department. 

Kind regards, 

on behalf of

Dr. Mary Diane Clark 

Academic Editor

PLOS ONE